**Molecular and seasonal characteristics of organic vapors in**
**urban Beijing: insights from Vocus-PTR measurements**
Zhaojin An[1,2], Rujing Yin[3], Xinyan Zhao[1], Xiaoxiao Li[4], Yuyang Li[1], Yi Yuan[1],
Junchen Guo[1], Yiqi Zhao[1], Xue Li[5], Dandan Li[1], Yaowei Li[2], Dongbin Wang[1],
Chao Yan[6], Kebin He[1], Douglas R. Worsnop[7,8], Frank N. Keutsch[2], Jingkun
Jiang[1,*]
[1]State Key Joint Laboratory of Environment Simulation and Pollution Control,
School of Environment, Tsinghua University, 100084, Beijing, China
[2]School of Engineering and Applied Sciences, Harvard University, Cambridge,
Massachusetts 02138, USA
[3]Key Laboratory of Industrial Ecology and Environmental Engineering (Ministry
of Education), School of Environmental Science and Technology, Dalian
University of Technology, 116024, Dalian, China
[4]School of Resource and Environmental Sciences, Wuhan University, 430072,
Wuhan, China
[5]School of Environment, Henan Normal University, 453007, Xinxiang, China
[6]Joint International Research Laboratory of Atmospheric and Earth System
Research, School of Atmospheric Sciences, Nanjing University, 210023,
Nanjing, China
[7]Institute for Atmospheric and earth System Research / Physics, Faculty of
Science, University of Helsinki, Helsinki 00014, Finland
[8]Aerodyne Research, Inc., Billerica, Massachusetts 01821, USA
*Corresponding author: Jingkun Jiang (email: jiangjk@tsinghua.edu.cn)

**Abstract**

Understanding the composition and evolution of atmospheric organic vapors is crucial for exploring their impact on air quality. However, the molecular and seasonal characteristics of organic vapors in urban areas, with complex anthropogenic emissions and high variability, remain inadequately understood. In this study, we conducted measurements in urban Beijing during 2021-2022 covering four seasons using an improved Proton Transfer Reaction-Mass Spectrometry (Vocus-PTR MS). During the measurement period, a total of 895 peaks were observed, and 512 of them can be assigned to formulas. The contribution of $C_xH_yO_z$ species is most significant, which composes up to 54% of the number and 74% of the mixing ratio of total organics. With enhanced sensitivity and mass resolution, various species with sub-ppt level or multiple oxygens (≥3) were observed, with 44% of the number measured at sub-ppt level and 31% of the number containing 3-8 oxygens. Organic vapors with multiple oxygens mainly consist of intermediate/semi-volatile compounds, and many of formulae detected were reported to be the oxidation products of various volatile organic precursors. In summer, the fast photooxidation process generate organic vapors with multiple oxygens and lead to an increase in both their mixing ratio and proportion. While in other seasons, the variations of organic vapors with multiple oxygens are closely correlated with those of organic vapors with 1-2 oxygens, which could be substantially influenced by primary emissions. Organic vapors with low oxygen content (≤ 2 oxygens) are comparable to the results obtained by traditional PTR-MS measurements in both urban Beijing and neighboring regions.

## 1 Introduction

Volatile organic compounds (VOCs) play a crucial role in the formation of ozone and fine particulate matter ($PM_{2.5}$) in the atmosphere, subsequently affecting air quality, climate, and human health (Carter, 1994; Williams and Koppmann, 2007; Jimenez et al., 2009; Hallquist et al., 2009). The sources and atmospheric evolution of VOCs in the atmosphere are complex due to the coexistence of compounds from primary emissions as well as secondary formation (Gentner et al., 2013; Gilman et al., 2015; Millet et al., 2015). Understanding their molecular characteristics is essential for studying their hydroxyl radial (OH) reactivities, ozone and secondary organic aerosol (SOA) formation potentials. However, the diverse range of species and wide distribution of oxidation products of atmospheric VOCs make it challenging to unravel their molecular properties (Goldstein and Galbally, 2007).

Instrumental advances have allowed for improving the understanding of the compositions and variations of VOCs at the molecular level, especially for oxygenated VOCs (OVOCs). Gas chromatography or multidimensional gas chromatography coupled with mass spectrometry is the most commonly used technology for VOC measurement, capable of detecting major non-methane hydrocarbons and select OVOCs (Lewis et al., 2000; Xu et al., 2003; Noziere et al., 2015). Proton Transfer Reaction-Mass Spectrometry (PTR-MS) enables real-time detection of VOCs without pre-concentration and separation, greatly enriching the molecular understanding of OVOCs due to its high sensitivity to oxygen-containing compounds (Hansel et al., 1995; De Gouw and Warneke, 2007; Yuan et al., 2017). Hundreds of OVOCs are detected and characterized in different areas using PTR-MS, e.g. urban (Wu et al., 2020), suburban (He et al., 2022), and forest areas (Pugliese et al., 2023). Recent developments in the ion-molecule reactor (IMR) configuration have greatly increased sensitivities and concurrently lowered the limits of detection of PTR-MS by several orders of magnitude by incorporating radio frequency electric fields to focus ions (Breitenlechner et al., 2017; Krechmer et al., 2018; Reinecke et al., 2023). A consequential issue is that these advanced PTR-MS typically need to eliminate lighter ions to protect the detector from overload, and similar to traditional PTR-MS, they are incapable of obtaining molecular structure information.

These improvements have expanded the detection capabilities of PTR-MS, particularly for organic vapors with lower volatility and multiple oxygens (≥3) (Riva et al., 2019), which enables the simultaneous measurement of VOC precursors and their primary, secondary, and higher-level oxidation products using a single instrument (Li et al., 2020). Despite their low concentrations, these vapors may condense on pre-existing aerosols and make a significant contribution to secondary aerosol growth and cloud condensation nuclei (Bianchi et al., 2019; Pospisilova et al., 2020; Nie et al., 2022). Organic vapors

with multiple oxygens are likely to be simultaneously detected by other chemical ionization mass spectrometry (CIMS), e.g., nitrate ($NO_3^-$), iodide ($I^-$), bromide ($Br^-$), and ammonium ($NH_4^+$) (Riva et al., 2019; Huang et al., 2021), which are widely used for measuring oxygenated organic compounds in the atmosphere (Bianchi et al., 2019; Ye et al., 2021; Huang et al., 2021). Therefore, using these improved PTR-MS can supplement our understanding of oxygenated organic vapors and facilitate the study of atmospheric chemical evolution of organics (Wang et al., 2020a).

The improved PTR-MS systems have gradually gained traction in research applications over the past few years, including measuring organics in controlled lab studies (Zaytsev et al., 2019a; Zaytsev et al., 2019b; Riva et al., 2019; Li et al., 2022a; Li et al., 2024a), emission sources (Sreeram et al., 2022; Yu et al., 2022; Yacovitch et al., 2023; Wohl et al., 2023; Jahn et al., 2023), and ambient air. For ambient measurements, observations in forested regions have been extensively conducted to study the compositions, variations, fluxes, and emissions of organics from different plants (Li et al., 2020; Li et al., 2021; Huang et al., 2021; Fischer et al., 2021; Thomas et al., 2022; Vettikkat et al., 2023; Vermeuel et al., 2023). Terpenes and their oxidation products with oxygen number up to 6 have been detected (Li et al., 2020). Diterpenes have been directly observed in the ambient air for the first time owing to the substantial improvement in sensitivity of Vocus-PTR (Li et al., 2020). Ambient measurement has been also conducted on a mountain in China, which found that terpenes and their oxidation products dominate the detected organic compounds, while the influence of industrial emissions can also be observed (Zhang et al., 2024).

In urban atmospheres, the sources and evolution of VOCs are considerably complex, potentially exhibiting distinct characteristics compared to forested areas. Several studies have carried out measurements in urban air using these improved PTR-MS. Jensen et al. (2023) conducted a one-month observation to address the production of reliable measurements. Coggon et al. (2024) evaluated the fragmentation and interferences of a series of urban VOCs. Pfannerstill et al. (2023 and 2024) measured hundreds of VOCs to calculate their emission fluxes in Los Angeles. A few low-signal species including dimethylamine, icosanal, dimethyl disulfide, and siloxanes emitted from diverse emission sources have been detected as a result of the enhanced sensitivity (Wang et al., 2020b; Chang et al., 2022; Jensen et al., 2023). However, the understanding of organic vapors with multiple oxygens in urban air, including their species, mixing ratios, diurnal profiles, and seasonal variations, remains inadequate.

In this study, we conducted measurements of organic vapors using a Vocus-PTR in urban Beijing during 2021-2022, covering four seasons. We present general characteristics of measured organic vapors and compare them with

traditional PTR-MS and previous Vocus-PTR measurements. We focus on
organic vapors with multiple oxygens (three or more), which have rarely been
individually analyzed in previous studies due to their low mixing ratios. Their
chemical compositions, atmospheric mixing ratios, diurnal and seasonal
variations are reported. Cluster analysis is further conducted to resolve the
main driving factors of their variations.

## 142 **2 Methods**

### 143 **2.1 Measurements**

The observation site is located in the central area of Tsinghua University, Beijing
(40°0'N, 116°20'E). It is an urban site with no significant direct influence from
industrial activities or heavy-traffic arteries (Fig. S1 in the supporting information,
SI). Details of this site can be found in the previous study (Cai and Jiang, 2017).
Organic vapors were measured by a Vocus-2R PTR-TOF-MS (Tofwerk AG and
Aerodyne Research Inc., referred to as Vocus-PTR hereinafter), which is
situated on top of a fourth-floor tower building, with its sampling inlet positioned
approximately 20 meters above the ground. The observation period is from May
1$^{st}$, 2021 to March 10$^{th}$, 2022, covering four seasons. Detailed information about
observation periods and their corresponding seasons is shown in Table S1.
The operating parameters of the Vocus-PTR used in this study are briefly
described here. In PTR-MS, VOCs are ionized via proton transfer by hydronium
ions ($H_3O^+$) in the IMR (Hansel et al., 1995; Yuan et al., 2016). The sensitivity
can be quantified based on the proton transfer reaction rate while
simultaneously considering ion transmission, detector efficiency, etc. (Cappellin
et al., 2012; Jensen et al., 2023). The ion source was supplied with a water
vapor flow of 20 sccm. The IMR was operated at 100°C and 2 mbar with axial
voltage of 600 V and quadrupole amplitude voltage of 450 V. The IMR operating
parameters were optimized to minimize the formation of water clusters. Mass
spectra were collected from m/z 11 to m/z 398 with a time resolution of 5 s,
achieving a mass resolution ~10,000 for $C_7H_9^+$ throughout the measurement
period. Ambient air was sampled via a tetrafluoroethylene (PTFE) tube (1.35 m
long, 1/4-inch OD) at a flow rate of 3 LPM to reduce wall losses, with only 150
sccm flow entering the Vocus-PTR. The sampling tube was heated to 50 ± 5°C
during the measurement. A regularly replaced Teflon filter (every 7 days) was
used in front of the sampling line to prevent the orifice from clogging. The data
within 30 minutes after membrane replacement was excluded. Measurements
were made on a 2-hour cycle with 110 min for ambient air, 5 min for zero gas,
and 5 min for fast calibration. The fast calibrations involved the use of mixed
calibration gases, with detailed information available in Table S2.
The ambient PM$_{2.5}$, NO$_2$, and O$_3$ data are from a state-operated air quality

station (Wanliu station), located approximately 3.6 km away from our observation site. The meteorological parameters, including temperature (T), relative humidity (RH), wind speed, and wind direction are also from Wanliu station. The diurnal variations of $PM_{2.5}$, $O_3$, $NO_x$, RH, and T in four seasons are shown in Figure S2.

## 2.2 Data processing

Data analysis of Vocus-PTR mass spectra, including mass calibration, baseline subtraction, and high-resolution peak fitting was conducted using Tofware (v3.2.3, Tofwerk AG and Aerodyne Research Inc.) within the Igor Pro 8 platform (WaveMetrics, OR, USA). The ambient mass spectra were averaged over 1 min for subsequent processing in Tofware. The peaklist used for high-resolution peak fitting was manually made based on mass spectra of both clean days ($PM_{2.5}$ < 75 $\mu g/m^3$) and polluted days ($PM_{2.5}$ $\geq$ 75 $\mu g/m^3$). The maximum mass error allowed for identifying peaks is 5-10 ppm, which is consistent of the error of mass calibration. When there are multiple options of formulas meeting the error limit under, especially at high molecular weights, a peak with oxygen numbers ≤ 8 and carbon numbers ≤ 20, and lower degree of unsaturation were selected; otherwise, the peak would be classified as unknown peak. The maximum peak area residual for each unit mass resolution is 5%. Subsequent analysis was performed in MATLAB R2022a (The MathWorks Inc., USA).

In PTR-MS, the sensitivities of organic vapors are typically determined through their direct linear correlation with their PTR rate constant ($k_{PTR}$). Vocus-PTR utilizes a big segmented quadrupole with a high-pass band filter, which detects ions < 35 m/z with reduced transmission efficiency (Krechmer et al., 2018). Consequently, determining sensitivities in Vocus-PTR involves consideration of both reaction efficiency and transmission efficiency. Figure S3a shows the measured sensitivities of mixed calibration gases and their corresponding $k_{PTR}$ values. The linear regression between $k_{PTR}$ and sensitivities was obtained based on sensitivities of $C_7H_9^+$, $C_8H_{11}^+$, $C_9H_{13}^+$, $C_{10}H_9^+$, and $C_5H_9O_2^+$ with an $R^2$ of 0.87. Sensitivities of other ions in mixed calibration gases may be influenced by transmission (ions labeled as gray) and fragmentation ($C_5H_9^+$, $C_{10}H_{17}^+$ and $C_{11}H_{11}^+$). The transmission efficiency of mixed calibration gases was calculated using sensitivities of mixed calibration gases, as shown in Figure S3b. The transmission efficiency of mixed calibration gases aligns well with the fitted transmission efficiency curve, except for $C_5H_9^+$, $C_{10}H_{17}^+$ and $C_{11}H_{11}^+$, which potentially experience fragmentation (fragmentation of measured ions are discussed below). For organic vapors without standards, their theoretical $k_{PTR}$ were used to constrain sensitivities, while for organic vapors with no theoretical $k_{PTR}$, an average $k_{PTR}$ of known species, $2.5×10^{-9}$ $cm^3$ molecule$^{-1}$ s$^{-1}$ was used to constrain their sensitivities. The theoretical $k_{PTR}$ of organic vapors are from previous studies (Zhao and Zhang, 2004; Cappellin et al., 2012; Sekimoto et

al., 2017). Average limits of detection (LODs, 1 min) of the measured compounds were determined using zero-gas background measurements taken every 2 hours during the observation periods, as shown in Figure S4. The LODs were calculated as 3 times the standard deviation of the zero-gas background divided by the obtained sensitivity. The LODs show a correlation with masses; as masses increase, instrument backgrounds decrease, leading to lower LODs. This trend was observed for species with different oxygen content, with LODs around $0.03 \pm 0.03$ pptv at m/z 200. Note that LODs in this study are one-minute averages, with raw 1-second data averaged to 1 minute before Tofware analysis as mentioned before, which may account for the lower LODs compared to those in Jensen et al. (2023). Data below the LODs were excluded from further analysis.

The fragmentation, water cluster, and interferences for calibrated and uncalibrated species were corrected. The ratio of the electric field strength (E) to the buffer gas number density (N) used in our study was 146.9 Td, and the gradient between BSQ skimmer 1 and skimmer 2 was 9.8 V, which in case limited the formation of water clusters, promoted the simple reaction kinetics, and improved the sensitivity, but may lead to stronger fragmentation. For $\alpha$-pinene, we identified its fragments based on GC chromatograms. The Vocus-PTR was calibrated in GC mode before atmospheric measurement. A total of 4 species were tested in GC mode, including severely fragmented $\alpha$-pinene. The spectrum of $\alpha$-pinene showed that the main fragment was $C_6H_9^+$. Several long-chain aldehydes and cycloalkanes may fragment on $C_5H_8H^+$, the ion typically attributed to isoprene in PTR-MS (Gueneron et al., 2015; Pfannerstill et al., 2023a; Coggon et al., 2024). We corrected isoprene signals following an approach by Coggon et al. (2024). The correction was calculated as follows:

$$\text{m/z } 69.07_{\text{Corrected}} = S_{69.07} - S_{111.12+125.13} \cdot f_{69.07/(111.12+125.13)} \quad (1)$$

$S_{69.07}$ is the signal measured at $C_5H_9^+$. $S_{111.12+125.13}$ is the signal of the isoprene interferences, referring to $C_8H_{15}^+$ (m/z 111.12) and $C_9H_{17}^+$ (m/z 125.13), which are dehydrated products from octanal and nonanal, respectively. $f_{69.07/(111.12+125.13)}$ was determined from nighttime data (0:00-4:00) of each period. Similarly, acetaldehyde was corrected for ethanol fragments. We also checked the fragments and water cluster list in Pfannerstill et al. (2023a) and Jensen et al. (2023). When the Pearson correlation coefficient r is greater than 0.95, the ions were considered as fragments or water clusters of the parent ion. We also tried to exclude the effects of unknown fragments and water clusters based on correlations of times series. Similar to Pfannerstill et al. (2023a), any ion showing a correlation with another ion with $r^2 > 0.97$ (if chemical reasonable) was analyzed for possible water clustering or fragmentation effects and added up with its parent ion. The ions corrected are listed as follows: $C_2H_4N^+$ with water cluster $C_2H_6NO^+$, $C_3H_7O^+$ with water cluster $C_3H_9O_2^+$, $C_5H_9^+$ with fragment $C_5H_7^+$, $C_7H_9^+$ with fragment $C_7H_7^+$, $CH_4NO^+$ with water cluster $CH_6NO_2^+$,

$C_2H_7O^+$ with water cluster $C_2H_9O_2^+$, $C_3H_3O_2^+$ with water cluster $C_3H_5O_3^+$,
$C_4H_5O_2^+$ with water cluster $C_4H_7O_3^+$, $C_3H_5^+$ with fragment $C_3H_3^+$, $C_2H_5O^+$ with
water cluster $C_2H_7O_2^+$, $C_2H_4NO^+$ with water cluster $C_2H_6NO_2^+$, $C_4H_5O_2^+$ with
water cluster $C_4H_7O_3^+$, $C_3H_3O_3^+$ with water cluster $C_3H_5O_4^+$, $C_6H_6NO^+$ with
water cluster $C_6H_8NO_2^+$, $C_8H_8NO_2^+$ with water cluster $C_8H_{10}NO_3^+$, $C_{10}H_{21}O^+$ with
water cluster $C_{10}H_{23}O_2^+$, $C_9H_{13}O_3^+$ with water cluster $C_9H_{15}O_4^+$, $C_{10}H_{13}O_3^+$ with
water cluster $C_{10}H_{15}O_4^+$, and $C_{14}H_{13}^+$ with water cluster $C_{14}H_{15}O^+$.
Here, we discuss the uncertainties of quantification for calibrated and
uncalibrated compounds. The uncertainty of calibrated ions ranges from 2% to
16% determined from the standard deviations of the fast calibrations during the
measurement periods. The semi-quantification was conducted for uncalibrated
compounds with their sensitivities constrained by $k_{PTR}$ linear relationship and
transmission efficiency. The uncertainty of these uncalibrated compounds
arising from linear fitting and transmission efficiency fitting is 20% using Monte
Carlo simulation. Additionally, undetermined fragmentations and water clusters
also contribute to the uncertainty, though we identified some potential
fragments and water clusters through the strength of correlations as previously
indicated. We acknowledge that this method cannot identify all fragments and
clusters, and fragments and clusters may still be present in the measured VOCs
and OVOCs. Further research is needed to explore the impact of fragments
and clusters on the measurements, particularly concerning OVOCs with
multiple oxygens.
Double bond equivalent (DBE), carbon oxidation state ($\overline{OS_C}$), and volatility of
organic vapors were calculated to address the chemical and physical properties
of detected organic vapors (see Text S1). The condensational growth rates
contributed by detected organic vapors were simulated using a kinetic
partitioning method, as detailed in Li et al. (2024b). For comparison, the
condensational growth rates of low volatile and extremely low volatile organic
compounds measured by nitrate-CIMS were also simulated (Li et al., 2024b).
The OH reactivities of detected organic vapors were calculated, and the rate
constants are from Data S1 in Pfannerstill et al. (2024) and Table S4 in Wu et
al. (2020). For species with unreported rate constants, we calculated the OH
reactivities for hydrocarbons and OVOCs using the reported median rate
constants of hydrocarbons and OVOCs, respectively.
Quantified or semi-quantified mixing ratios were further processed by cluster
analysis to investigate their characteristics. Intraclass correlation coefficient
(ICC) is a suitable method for assessing the consistency of trends in
unbalanced data. It quantifies the stability of differences between two sets of
measurement results, enabling evaluation of their consistency. ICC combined
with k-means cluster analysis were used. ICC(C, 1) was selected among
several typical consistency evaluation parameters for its evaluation results
exhibit the highest level of differentiation based on factual evidence (Qiao et al.,
2021). ICC(C, 1) was calculated as follows:

$$ICC(C, 1) = (D(X + Y) - D(X - Y))/(D(X + Y) + D(X - Y)) \qquad (1)$$

where $D(\cdot)$ is the arithmetic operators of variance. $X$ and $Y$ are two sets of
measurement data, in this case referring to the mixing ratios of any organic
vapors we are concerned about. The ICC matrices of various organic vapors
were subsequently utilized as input for k-means analysis. Square Euclidean
distance was selected to calculate the distances between different organic
vapors.

## 3 Results and discussion

### 3.1 General characteristics of organic vapors

During the measurement period, a total of 895 peaks were observed, and 512
of them can be assigned to formulae, divided into $C_xH_y$, $C_xH_yO_z$, $C_xH_yN_i$, and
$C_xH_yO_zN_i$ categories based on their elemental compositions (Fig. 1a). $C_xH_yO_z$
composes up to 54% of the total number of formulae followed by $C_xH_yO_zN_i$,
$C_xH_y$, and $C_xH_yN_i$, with proportions of 26%, 14%, and 6%, respectively (Fig. 1b).
$C_xH_yO_z$ dominates contributing 74% of the annual median mixing ratios of total
organics, followed by $C_xH_y$, $C_xH_yO_zN_i$, and $C_xH_yN_i$, with proportions of 22%, 2%,
and 2%, respectively (Fig. 1c). In addition to these resolved formulae, we also
detect 18 peaks containing other elements such as S, Cl, Si, etc., and 79
CH(O)(N) peaks that do not comply with nitrogen rules, which we consider as
fragments or free radicals. Others are unknown peaks for which formulae
cannot be assigned or water clusters/fragments excluded from analysis. The
mixing ratios of organic vapors vary substantially in urban Beijing, ranging from
0.01 parts per trillion (ppt) to 10 parts per billion (ppb) in volume under a time
resolution of 1 min, with many species detected at sub-ppt levels notably (Fig.
1d). The units of the mixing ratio in the following text are all volume fractions.
As the molecular masses of organics increase, their annual median mixing
ratios decrease. The mixing ratios of $C_xH_yO_z$ and $C_xH_yO_zN_i$ categories start to
decrease below the ppt level above molecular weights of 160 and 125,
respectively.
With enhanced sensitivity and mass resolution, an increased number of
formulae have been identified compared to traditional PTR-MS measurements
in urban Beijing, especially formulae with lower mixing ratios and higher oxygen
contents. Note that most organics with low mixing ratios have high oxygen
content. 44% number of formulae measured in this study are at sub-ppt level
while 31% number of formulae are between 1 and 10 ppt (Fig. 1e). Only
compounds detected above ppt levels were previously reported in urban sites
within Beijing (Sheng et al., 2018; Li et al., 2019), as well as at a suburban site
located 100 km southwest of Beijing (He et al., 2022). Simultaneously, organic
vapors with multiple oxygens ($C_xH_yO_{\geq 3}$ and $C_xH_yO_{\geq 3}N_i$ species) have been
successfully detected in this study in the urban atmosphere. Traditionally, they
have been often recognized as total $C_xH_yO_{\geq 3}$ species, with no individual
analysis in traditional PTR-MS (Yuan et al., 2023; Li et al., 2022b; He et al.,
2022). Many other studies only focus on reporting OVOCs containing up to 2-3
oxygens or omit to address the presence of nitrogen containing OVOCs (Wang
et al., 2021a; Liu et al., 2022). The low mixing ratios and high wall losses of
organic vapors with multiple oxygens impact the detection in traditional PTR-
MS (Breitenlechner et al., 2017). Figure 2a reinterprets the mass defect plot of
measured organics with a focus on oxygen numbers, ranging from 0 to 8. The
analysis of mixing ratio levels and variations of organic vapors with multiple
oxygens (≥3) are shown in Section 3.2. Organic vapors with low oxygen content
(≤2) are reported in Section 3.3. Subsequent comparison of Vocus-PTR and
traditional PTR in urban Beijing and both Vocus-PTR measurements in urban
Beijing and European forests are also shown in Section 3.3.

## 3.2 Organic vapors with high oxygen content

195 observed organics with multiple oxygen atoms account for 38% in number
of the total organics, including 136 species of $C_xH_yO_{\geq 3}$ and 59 species of
$C_xH_yO_{\geq 3}N_i$. Organics with oxygen numbers 3 and 4 dominates within the
$C_xH_yO_{\geq 3}$ and $C_xH_yO_{\geq 3}N_i$ species (Fig. 2b and Fig. 2c). Organics with oxygen
number of 3, 4, 5, and ≥6 comprise 15%, 11%, 7%, and 6% of the total species
number of $C_xH_yO_z$ compounds, respectively. While compounds with oxygen
number of 3, 4, 5, and ≥6 comprise 15%, 12%, 7%, and 2% of the total species
number of $C_xH_yO_zN_i$ compounds, respectively.
The measured organic vapors with multiple oxygens are mainly intermediate
volatile organic compounds (IVOCs) and semi-volatile organic compounds
(SVOCs). The dominant carbon numbers range from 5 to 9 and DBE between
1-5, accounting for over three-quarters of the total species number of organic
vapors with multiple oxygens (Fig. 3a and Fig. 3b). The maximum occurrence
of organic vapors with 3 or 4 oxygen atoms is observed within the carbon range
of 7-8 and a DBE value of 2. For organic species with 5 or more oxygens, they
reach their peak at a smaller carbon number of 4-5 and a higher DBE value of
3. Aromatic VOCs have DBE values no smaller than 4, while aliphatic VOCs
usually have DBE values smaller than 2. For organic vapors with DBE between
2-3, they are likely oxidation products of aliphatic and aromatic VOCs (Wang et
al., 2021b; Nie et al., 2022). For the same number of carbon atoms, organic
vapors with a higher number of oxygen atoms exhibit a higher carbon oxidation
state (as shown in Figure S5). Compared to organic vapors with 3 or 4 oxygen
atoms, organic vapors with 5 or more oxygens have undergone more extensive
atmospheric oxidation and functionalization processes (Kroll et al., 2011;
Isaacman-Vanwertz et al., 2018). Based on calculated volatility, 81% of the

species are IVOCs, and the remaining 19% are SVOCs (Fig. 3c). With the increase in oxygen number, the volatility of the compounds gradually decreases, while the potential partitioning to aerosols increases, manifested by a gradual reduction in the peak values of the $log_{10}C_0$. Compounds containing nitrogen, referred to shaded bars with white stripes in Figure 3c, have a lower volatility compared to non-nitrogen species.

The annual median mixing ratio of measured organic vapors with multiple oxygens in median ± standard deviation is 2.0 ppb ± 1.0 ppb, accounting for 4% of the total $C_xH_yO_z$ and $C_xH_yO_zN_i$ mixing ratios. For $C_xH_yO_z$ category, the annual median mixing ratios of species with 3, 4, 5, and ≥6 oxygens are 1.4 ppb, 186.4 ppt, 18.1 ppt, and 6.4 ppt, respectively. For $C_xH_yO_zN_i$ category, the annual median mixing ratios of species with 3, 4, 5, and ≥6 oxygens are 49.9, 24.5, 2.6, and 0.5 ppt, respectively (Fig. 2d and 2e). Organic vapors with 3 oxygens constitute the overwhelming majority of the mixing ratio of measured organic vapors with more than three oxygens. As a result, the mixing ratio-weighted carbon number and DBE distributions (Fig. 3d and Fig. 3e) are significantly different from that of species number distributions for organic vapors with multiple oxygens. The mixing ratios of species with carbon numbers ranging from 2 to 6 are significantly higher, with those containing four carbons exhibiting the highest mixing ratios. Similarly, the mixing ratios of species with DBE ranging from 0-4 are notably higher than that of other DBE values. As compounds containing 3 oxygens dominate the mixing ratio, IVOCs nearly entirely contribute to the mixing ratio-weighted volatility of organic vapors with multiple oxygens (Fig. 3f). The mixing ratios of organic vapors with multiple oxygens measured in this study are higher than other studies, which will be detailed in Section 3.3.

Though the contribution of the measured IVOCs and SVOCs to the overall VOC mixing ratio is low, their contribution to the condensational growth rates is non-negligible, which may influence the growth of new particles (Ehn et al., 2014), SOA formation (Jimenez et al., 2009), and haze (Nie et al., 2022). The condensational growth rates of total organic vapors are calculated, including extremely low, low, and semi volatile organic compounds detected by nitrate-CIMS and I/SVOCs detected by Vocus-PTR. The contribution to the condensational growth rate from I/SVOCs detected by Vocus-PTR increases with particle size and decreases with temperature. For 8 nm particles, the contribution of SVOCs detected by Vocus-PTR is 9%, while IVOCs contribute 1%. For 40 nm particles, the contribution of SVOCs increases to 13%, and IVOCs rise to 4%. At sub-zero temperatures for 8 nm particles, the SVOC contribution detected by Vocus-PTR can reach up to 21%, with IVOCs contributing 10%.

The molecular formulae of the measured organic vapors with multiple oxygens are displayed in the mass spectra, categorized by carbon numbers ranging from

2-11 (Fig. 4 and Table S3). Many of the formulae are reported as oxidation products of various VOC precursors in previous studies. Take isoprene as an example, detected formulae are reported as various oxidation products of isoprene, including $C_5H_{10}O_3$ and subsequent oxidation products in C5 species, e.g., $C_5H_8O_6$, $C_5H_9NO_4$, etc. (Wennberg et al., 2018). For several C4 species, such as $C_4H_7NO_4$, $C_4H_4O_3$, etc., they are reported as oxidation products of two additional important oxidation products of isoprene, methacrolein (MACR) and methyl vinyl ketone (MVK). We also see formulae reported as oxidation products of precursors such as benzene (C6) (Priestley et al., 2021), alkyl-substituted benzenes (C7-C9) (Pan and Wang, 2014; Wang et al., 2020c; Cheng et al., 2021), and monoterpenes (C10) (Rolletter et al., 2019). Besides, we can also detect some organic vapors with relatively low DBE (≤3), which may originate from the oxidation of aliphatic precursors. For example, $C_5H_8O_4$ observed are reported as one of the oxidation products of C5 aldehyde, the photolysis of which release OH radicals. This mechanism may explain the source gap of OH radicals between simulations and observations in low nitrogen oxide and high VOCs regimes (Yang et al., 2024). Note that these species may be oxidation products as reported by previous studies; however, confirming this would require additional techniques such as GC.

Measured molecular formulae may react with OH radicals, contributing to OH reactivity. The calculated OH reactivity of organic vapors with multiple oxygens account for 6% of the total detected VOCs, with an average annual value of 1.2 $s^{-1}$. Previous studies show differences between measured and calculated or modeled OH reactivity (Hansen et al., 2014), and unmeasured species from photochemical oxidation likely explain this gap (Ferracci et al., 2018). Therefore, the OH reactivity contributed by detected organic vapors with multiple oxygens in this study may potentially reduce this gap, thereby improve the accuracy of diagnosis of sensitivity regimes for ozone formation (Wang et al., 2024). Using Vocus-PTR has the potential to simultaneously measure both precursors and multi-generational oxygenated products, which is beneficial for studying the evolution process of organic compounds in the atmosphere.

As for the seasonal variations, the overall mixing ratio of organic vapors with multiple oxygens is the highest in winter, followed by summer, spring and the lowest in autumn (Fig. 5a). The mixing ratios expressed in median ± standard deviation (ppb ± ppb) are 1.9 ± 0.5, 1.9 ± 0.9, 1.4 ± 1.2, and 2.2 ± 0.8 for spring, summer, autumn, and winter, respectively. Compounds with different oxygens exhibit different seasonal variations, shown in Figure 5b and 5c and Table S4. For $C_xH_yO_z$ with 3 or 4 oxygens, the mixing ratios are higher in winter than in other seasons, while for compounds containing 5 or more oxygens, the mixing ratios are highest in summer. For $C_xH_yO_zN_i$ with 3 or 4 oxygens, the mixing ratios are high in both summer and winter, while for compounds containing 5 or more oxygens, the mixing ratios are high in summer and spring. As the oxygen

number increases, the contribution from secondary sources becomes greater, and the high mixing ratio of oxidants in summer intensifies this process. Thus, the fraction of the mixing ratio of compounds with multiple oxygens increases with the oxygen number in summer (Fig. 5d). In winter, the mixing ratios of compounds containing five or more oxygens are substantially suppressed, which may be due to reduced generation. Alternatively, it could be that these compounds belong to SVOCs, with a majority being partitioned onto particulate matter at low temperatures.

The seasonal variations of organic vapors with multiple oxygens differ from those of total OVOCs (Fig. S6), with the latter's mixing ratio being primarily influenced by organic vapors containing 1-2 oxygen atoms. The mixing ratio of total OVOCs in winter is substantially higher than in the other three seasons, followed by autumn and summer, with the lowest mixing ratio observed in spring. The seasonal variations of OVOCs are partly caused by the variation of mixing layer height (Li et al., 2023), which is lowest in winter. Cluster analysis is performed to further explore the dominated driving factors of the seasonal variations of organic vapors with multiple oxygens. Three clusters are identified in each season based on the diurnal profiles of each compound. To increase the interpretability of the clusters, two of them are merged. Figure 6 and Figure S7 shows the cluster results for organic vapors with multiple oxygens. For comparison, cluster analysis is performed on organic vapors with 1-2 oxygens as well (Fig. S8 and Fig. S9).

Daytime clusters, where the peak occurs during the daytime, were identified across the four seasons for organic vapors with multiple oxygens (shown as cluster 1 in Fig. 6). Daytime clusters start to rise at 6:00-7:00 (6:00 for summer and 7:00 for other seasons), peak at 11:00-14:00 and then slowly decrease, following the diurnal variation of solar radiation (Li et al., 2023), ozone and temperature (Fig. S2). Figure S10 further demonstrates the dependence of daytime clusters on temperature. The mixing ratio of daytime clusters show an apparent increase in summer (when temperature is higher than 15 °C), which indicates that higher temperatures accompanied by an increase in solar radiation and ozone favors the formation of daytime clusters. The number and corresponding mixing ratios of species allocated to the daytime clusters vary in four seasons. In summer, the vast majority of species (77%) exhibit daytime characteristics, with a mixing ratio percentage as high as 85%, which may be related to the strongest solar radiation (Li et al., 2023) and lowest NOx concentrations (Fig. S2). The contribution of daytime clusters in autumn is also significant, with 67% and 58% of the species and mixing ratios being accounted for. The noon peaks of daytime clusters in winter and spring are relatively less pronounced, with the species and mixing ratio day/night ratios also being comparatively lower. The afternoon peak of daytime clusters in autumn and winter are accompanied by a decrease in mixing layer height (Li et al., 2023).

For organic vapors with 1 or 2 oxygens, a significant daytime cluster was
observed only in summer (Fig. S8 d-f).
Another cluster type is considered to be nighttime clusters, as the
corresponding species have their highest mixing ratios at night. Unlike the
daytime cluster, the diurnal variations of nighttime clusters are different in four
seasons (Fig. 6). In spring, the nighttime cluster comprises over 86% of
nighttime species and 77% of mixing ratios, and it peaks at 4:00 with low
daytime values. The nighttime clusters in winter and autumn show bimodal
diurnal variations, with the highest peak occurring during the night from 19:00
to 23:00, and the second peak appearing during the day from 8:00 to 12:00. 47%
and 33% of species exhibit the characteristics of the nighttime cluster in winter
and autumn, constituting 58% and 42% of the mixing ratio, respectively. The
contribution of the nighttime cluster is minimal in summer, reaching its peak at
midnight. We found that each nighttime cluster of organic vapors with multiple
oxygens shows good consistency with the corresponding major clusters of
organic vapors containing 1-2 oxygens (Fig. S8 and Fig. S11), while the mixing
ratios during midday differ. Nighttime clusters also show better consistency with
$PM_{2.5}$ compared to daytime clusters (Fig. S2), which may be related to mixed
sources.
Most organic vapors with multiple oxygens could be assigned to different
clusters in different seasons (Fig. S12). Only a small number of species can be
categorized into the same cluster in four seasons. Figure S13 shows the
average C, H, O, and N number of species assigned to daytime cluster 0-4
times during the four seasons. As compounds exhibit more characteristics
associated with daytime cluster, there is no significant change in the carbon
number, but there is an increase in hydrogen and oxygen number, and a
decrease in nitrogen number. This may be due to multi-step oxidation reactions
in the atmosphere, causing an increase in oxygen number and DBE of species
(Kroll et al., 2011; Isaacman-Vanwertz et al., 2018), with diurnal variations
peaking at noon as a result of the strongest photochemistry. The decreasing
trend of the number of nitrogen atoms in Figure S13 indicates that nitrogen
containing compounds measured in this study are more likely to come from
nocturnal production or emissions. Regarding the average elemental
composition (C, H, O, and N) of species assigned to two clusters (see Fig. S14),
daytime clusters typically exhibit higher oxygen content and lower H/C
compared to nighttime clusters, providing further evidence supporting the
atmospheric photochemical origin of daytime clusters. The nighttime clusters
have higher nitrogen contents than daytime clusters, indicating more of the
impacts of nocturnal sources.
**3.3 Organic vapors with low oxygen content**
In addition to multiple oxygens, organic vapors with low oxygen content were

also measured in urban Beijing in this study. Here we primarily discuss comparisons between the results of this study and those of previous studies. The mixing ratios and variations of typical VOCs measured in this study are comparable to the results obtained by traditional PTR-MS measurements in both urban Beijing and neighboring regions. Figure S15 shows the diurnal profiles of 12 representative VOCs in four seasons. OVOCs of $C_2H_4O$, $C_3H_6O$, and $C_4H_4O$, usually identified as acetaldehyde, acetone, and furan, are mainly from anthropogenic sources as reported by previous studies (Qian et al., 2019). Their diurnal variations exhibit a characteristic of being higher at night and lower during the day, similar to other studies reported in Beijing during the winter (Sheng et al., 2018; He et al., 2022). The mixing ratios of acetaldehyde, methyl ethyl ketone (MEK), and furan in winter are slightly lower than those observed in winter Beijing in 2016 and 2018 (Sheng et al., 2018; He et al., 2022). The winter mixing ratios of acetone are higher than other seasons and observed in other studies, indicating an unknown emission source during winter. The mixing ratios of benzene ($C_6H_6$), toluene ($C_7H_8$), and naphthalene ($C_{10}H_8$) in winter are slightly lower than reported in winter in Beijing during the past few years (Sheng et al., 2018; Li et al., 2019; He et al., 2022), possibly due to improvements in air pollution policies, especially those targeting emissions from residential combustion and motor vehicles (Liu et al., 2023). As for phenols, the mixing ratios of $C_6H_6O$ are similar to measurement at a background site in the North China Plain in winter, while the mixing ratios of $C_7H_8O$ are much lower than that (He et al., 2022). High mixing ratios of biogenic emissions in summer are observed, for example isoprene ($C_5H_8$) and the sum of its oxidation products MACR and MVK (Apel et al., 2002) have peak mixing ratios of 2.6 ppb and 0.6 ppb, respectively. Their mixing ratios in winter are lower and consistent with other studies (Sheng et al., 2018; He et al., 2022).

The mixing ratio fractions of organic categories in urban Beijing using Vocus-PTR differ from the results obtained using traditional PTR-MS. Previous studies in Beijing have only reported a few selected VOCs up to around 100 species, resulting in limited results on systematic characterizations of VOCs using PTR-MS in Beijing (Sheng et al., 2018; Li et al., 2019; Wang et al., 2021a; Liu et al., 2022). Therefore, we compare with a suburban site, Gucheng, which is located 100 km southwest from our site. The two sites (urban Beijing and Gucheng) are both located in the North China Plain and are subject to regional air pollutions simultaneously. Figure S16 shows the comparison results of five categories, including $C_xH_y$, $C_xH_yO$, $C_xH_yO_2$, $C_xH_yO_{\geq 3}$, and N/S containing compounds. The first difference is that the mixing ratio fraction of species containing two or more oxygens measured by Vocus-PTR is higher than those measured by traditional PTR-MS. The mixing ratio fractions of $C_xH_yO_2$ and $C_xH_yO_{\geq 3}$ in Vocus-PTR are 12% and 4%, respectively, whereas they are 6% and 1% for traditional PTR-MS. In terms of mixing ratios, the mixing ratio of $C_xH_yO_{\geq 3}$ is approximately double in Vocus-PTR compared to traditional PTR-MS, while the mixing ratio of

$C_xH_yO$ is half compared to traditional PTR-MS measurement. The mixing ratio
of $C_xH_yO_2$ remains similar. This is because Vocus-PTR can detect more OVOCs
with multiple oxygens due to its high sensitivity and mass resolution, whereas
due to its low transmission efficiency for low masses, it is difficult to detect high
mixing ratio OVOCs such as methanol and formaldehyde. The other difference
is that the mixing ratio and the corresponding fraction of $C_xH_y$ species measured
by Vocus-PTR are much lower than those measured by traditional PTR. For
several major $C_xH_y$ compounds such as benzene, C7, C8, and C9 aromatics,
their mixing ratios are comparable between the two methods. The main
difference between the two methods lies in the mixing ratio of low-mass
hydrocarbons. Overall, when applied to the urban atmosphere, Vocus-PTR has
advantages in measuring oxygenated VOCs, especially with multiple oxygens.
However, it has limitations in measuring low molecular weight VOCs due to the
low-mass cutoff in the transmission efficiency.
The molecular characteristics of organic vapors measured by Vocus-PTR in
urban Beijing show several differences from those in forested areas (Li et al.,
2020; Huang et al., 2021; Li et al., 2021). Firstly, organics up to 300 m/z can be
observed in forested areas, while organics up to 230 m/z are observed (Fig.
1a). Two main reasons are responsible for this. The complexity of the species
introduces challenges in interpreting mass spectra, which is evidenced by the
total number of species being similar to existing atmospheric measurements
using Vocus-PTR, despite a narrower mass range in this study. The higher
particulate matter concentrations in urban areas provide a larger sink for
organic vapors (Deng et al., 2020), and this loss effect is especially pronounced
for compounds with high molecular weights due to their lower volatility. The
second difference is that, $C_xH_yO_z$ and $C_xH_yO_zN_i$ species are the dominant
organics in both urban and forested areas, whilst $C_xH_yN_i$ species are more
common and abundant in urban areas, which may come from biomass burning
emissions (Laskin et al., 2009). Thirdly, VOCs with low carbon and oxygen
number play a more significant role in total organic mixing ratio compared to
results from forested regions. As shown in Figure S17a, $C_2$ and $C_3$ organics
contribute 79% of the total organic mixing ratio in this study, while $C_4$-$C_6$
organics contribute approximately 75% in forested regions. In contrast to
forested areas, where VOCs and IVOCs mixing ratios are comparable, the
majority of the total organic mixing ratio is attributed to VOCs in this study (Fig.
S17b). Typical $C_2$ and $C_3$ organics, such as $C_3H_6O$, $C_2H_4O$, and $C_2H_4O_2$,
contribute 14%, 11%, and 5%, respectively, to the total organic mixing ratio,
which are mainly originated from anthropogenic emissions including industrial
and vehicular activities, solvent utilization, and other sources (Qian et al., 2019).

## 4 Conclusions

In this study, we explore the molecular and seasonal characteristics of organic vapors in urban Beijing using a Vocus-PTR over four seasons. A total of 895 peaks are observed, and 512 of them can be assigned to formulae. The contribution of $C_xH_yO_z$ species is most significant, which compose up to 54% of the number and 74% of the mixing ratios of total organics. With enhanced sensitivity and mass resolution, an increased number of species were observed compared to traditional PTR-MS measurements in urban Beijing, especially compounds with lower mixing ratios and higher oxygen content. 44% species in number measured in this study are at sub-ppt level and 31% species in number contain 3-8 oxygens, resulting in a higher fraction of species containing three or more oxygens compared to traditional PTR-MS measurements. Organic vapors with low oxygen content are comparable to those obtained in both urban Beijing and neighboring regions, and they exert a more substantial influence on the overall organic mixing in forested areas.

The mixing ratio of organic vapors with multiple oxygens accounts for 4% of the total VOC mixing ratio, with the highest levels observed in winter, followed by summer, spring, and the lowest in autumn. These vapors also make a non-negligible contribution to condensational growth and OH reactivity. In summer, the majority of species are aligned to daytime cluster (peaking at noon), primarily originating from the photooxidation process. As the oxygen number increases, the impact of the photooxidation process becomes more pronounced, leading to an increase in both mixing ratio and proportion of organic vapors with multiple oxygens during summer. In spring and winter when the nighttime cluster (peaking at night) dominated, the variations of organic vapors with multiple oxygens are strongly correlated with organic vapors with one or two oxygens. The measured compositions and seasonal variabilities of organic vapors with multiple oxygens emphasize the importance of high sensitivity and high mass resolution measurements in urban atmosphere, suggesting prospective for future research.

**Data availability**

Data are available upon request from the corresponding author.

**Supporting Information**

The content of the SI includes the map of the observation site (Fig. S1); the diurnal variations of $PM_{2.5}$, $O_3$, $NO_x$, RH, and T in four seasons (Fig. S2); calibration results of mixed calibration gases (Fig. S3); average limits of detection (1 min) for detected compounds (Fig. S4); carbon oxidation state of

organic vapors with different oxygens (Fig. S5); boxplot of total OVOC mixing ratios in four seasons (Fig. S6); diurnal variation cluster results of organic vapors with multiple oxygens (Fig. S7); cluster results of organic vapors with one or two oxygens (Fig. S8-S9); dependence of daytime clusters on temperature (Fig. S10); dependence of nighttime clusters on major clusters of organic vapors with 1-2 oxygens (Fig. S11); the distribution of organic vapors with multiple oxygens across different clusters (Fig. S12); average C, H, O, and N number of organic vapors containing multiple oxygens with different diurnal patterns (Fig. S13); average C, H, O, and N number of organic vapors containing multiple oxygens in two clusters (Fig. S14); diurnal profiles of representative VOCs in four seasons (Fig. S15); comparison results with Gucheng site (Fig. S16); molecular characteristics of total measured organic vapors by Vocus-PTR (Fig. S17); the observation periods of Vocus-PTR (Table S1); information about calibration gases (Table S2); main $C_xH_yO_{\geq 3}$ and $C_xH_yO_{\geq 3}N$ species measured in this study (Table S3), and seasonal mixing ratios of OVOCs with multiple oxygens (Table S4).

## Author contributions

Conceptualization: JJ and ZA. Data collection and analysis: ZA, RY, XZ, XxL, YY, JG, YuL, YZ, and XuL. Writing-original draft: ZA. Writing-review and editing: XxL, DL, YaL, DW, CY, KH, DRW, FNK, and JJ.

## Competing interests

At least one of the (co-)authors is a member of the editorial board of *Atmospheric Chemistry and Physics*.

## Financial support

This work has been supported by the National Natural Science Foundation of China (Grant NO. 22206097, 22188102, and 22106083) and Samsung PM$_{2.5}$ SRP.

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

1112

1113

 **Figures**

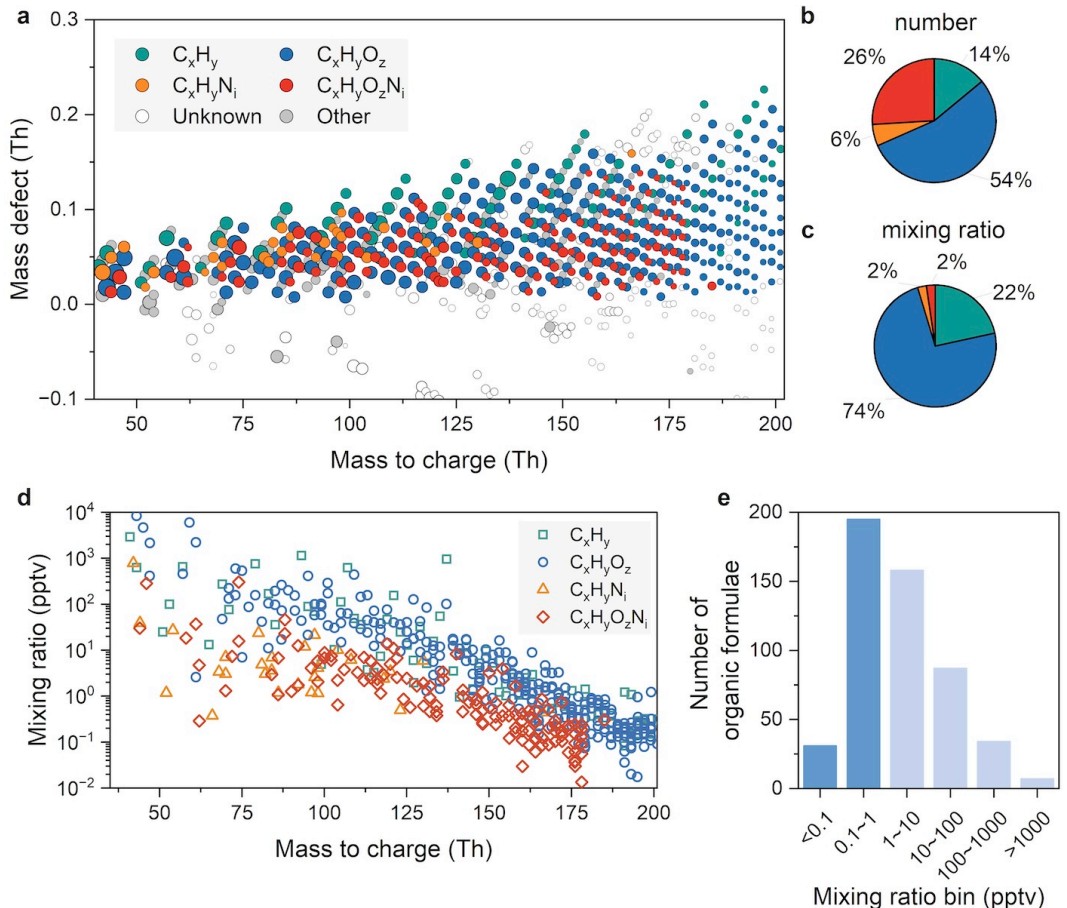

1115

Figure 1. Identified formulae in urban Beijing using Vocus-PTR. (a) Mass defect plot. The sizes of the bubbles represent the annual median mixing ratios. The bubbles are colored by different elemental compositions as labeled in the legend. The "unknown" refers to fitted peaks without matched formula. The "other" refers to peaks containing elements other than C, H, O, and N or fragment peaks (or radicals). (b) Pie chart of the number of identified formulae. (c) Pie chart of the annual median mixing ratios of identified formulae. The color scheme of the pie charts is the same to that of the mass defect plot. (d) The annual median mixing ratios of identified formulae versus their masses. (e) Histogram of annual mixing ratios of identified formulae. Bins with values less than 1 ppt are emphasized in dark blue color.

1127
1128

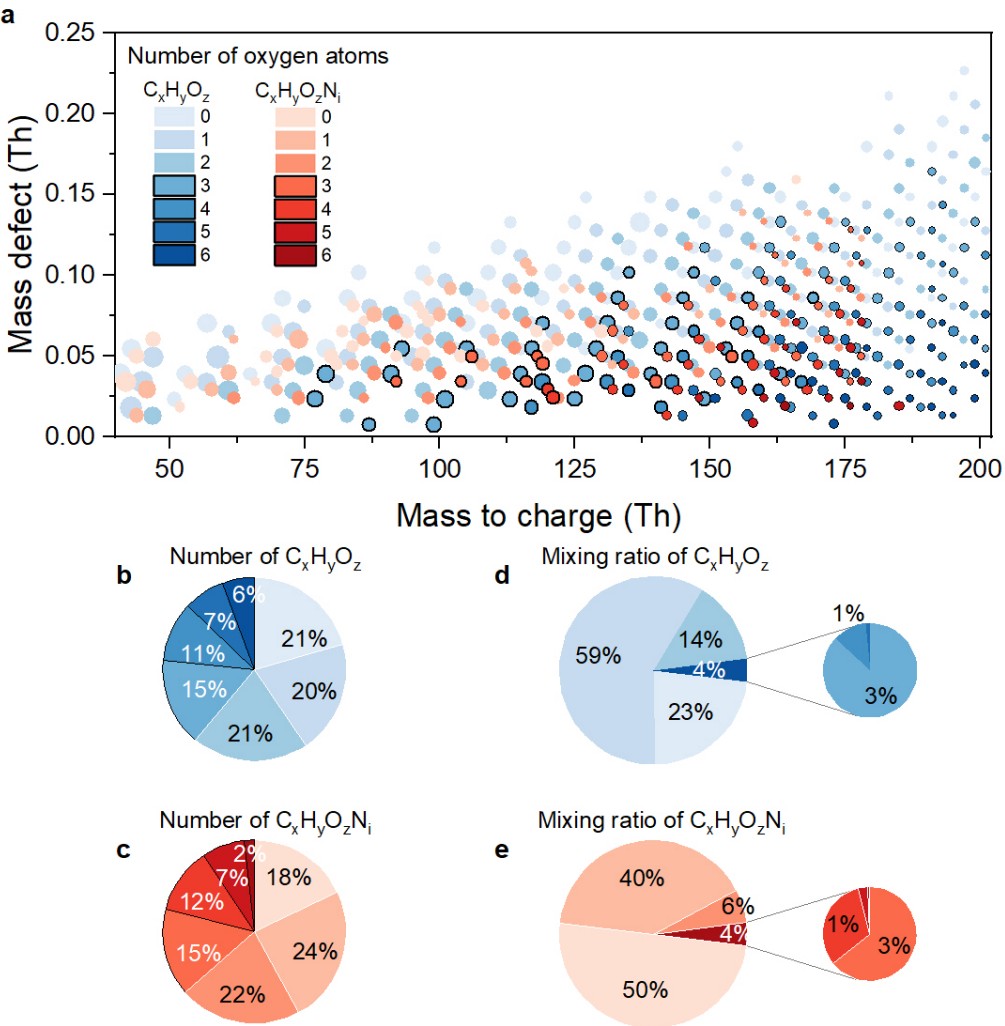

1129

Figure 2. Organic vapors of different oxygen content. (a) Mass defect plot. The sizes of the bubbles represent the annual median mixing ratios. The bubbles are colored by different oxygen numbers as labeled in the legend. Bubbles representing organic vapors with 3 or more oxygens are highlighted with black borders. Bars labeled as 6 refers to organic vapors with oxygen number equal or larger than 6. (b) Pie chart of the number of $C_xH_yO_z$ species. (c) Pie chart of the number of $C_xH_yO_zN_i$ species. (d) Pie chart of the mixing ratio of $C_xH_yO_z$ species. (e) Pie chart of the mixing ratio of $C_xH_yO_zN_i$ species. The color scheme of the pie charts is the same to that of the mass defect plot.



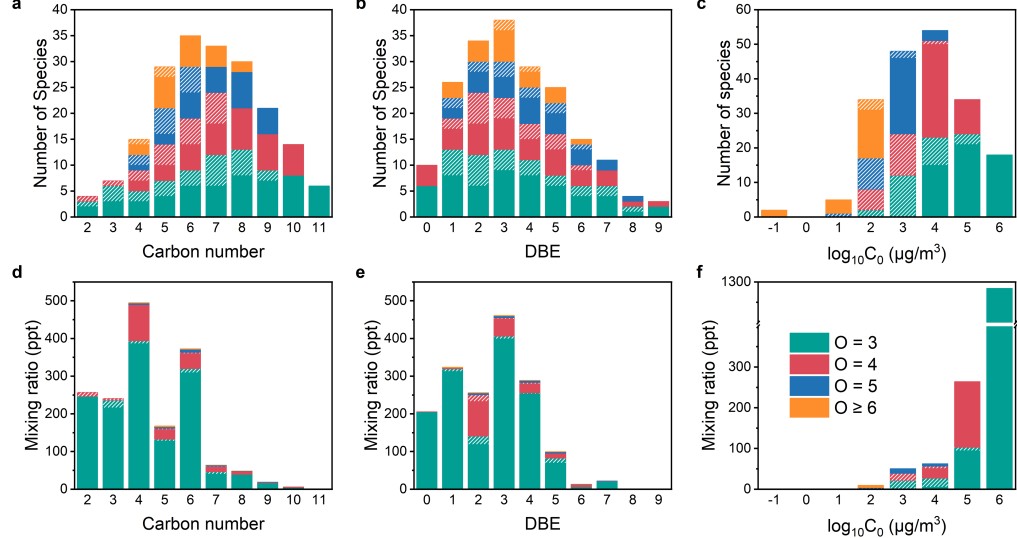


Figure 3. Distribution of carbon number, double bond equivalent (DBE), and
volatility of organic vapors with multiple oxygens. Panels (a) - (c) represent
species number distributions of carbon number, DBE, and volatility, respectively.
Panels (d) - (e) represent mixing ratio distributions of carbon number, DBE, and
volatility, respectively. Different color of bars refers to compounds with different
oxygen content. Bars without white stripes represent $C_xH_yO_{\geq 3}$, while shaded
bars with white stripes represent $C_xH_yO_{\geq 3}N$. Y axes refer to annual median
mixing ratios.

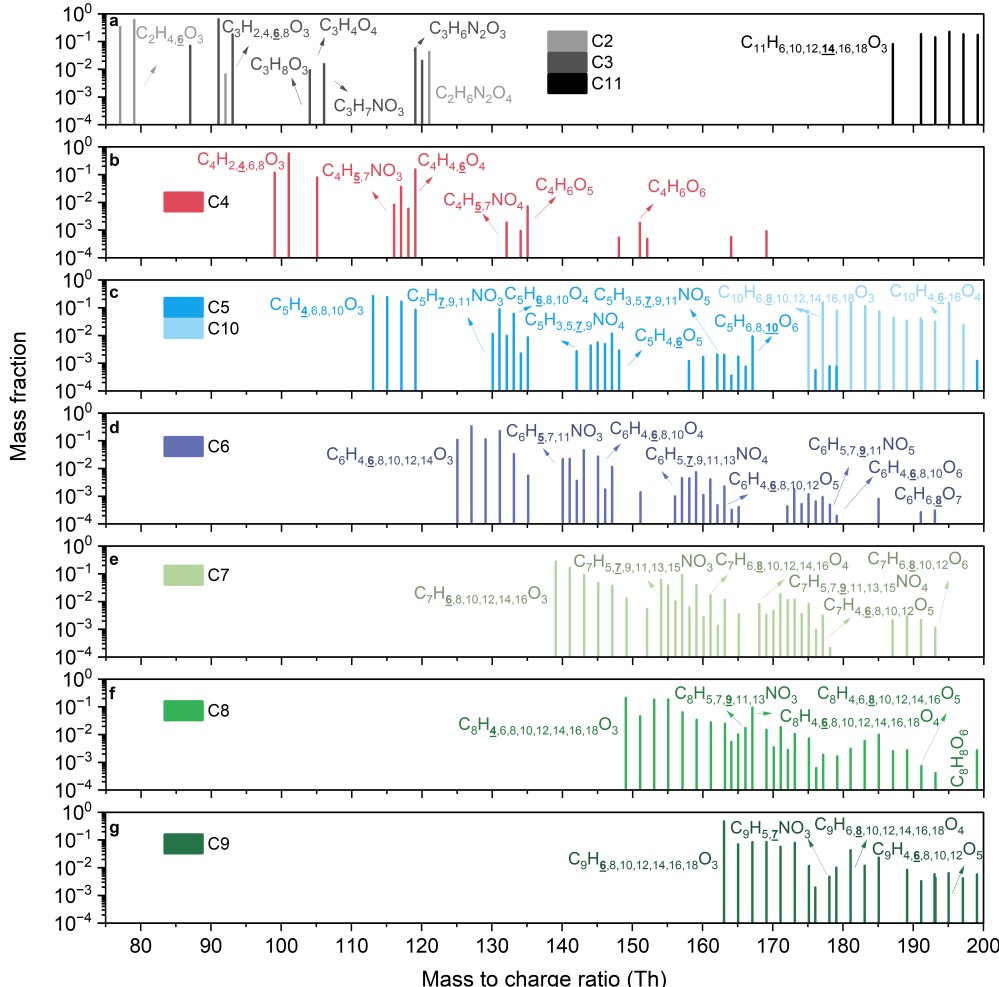

Figure 4. Mass spectra of organic vapors with multiple oxygens with different carbon numbers: (a) C2, C3, and C11; (b) C4; (c) C5 and C10; (d) C6; (e) C7; (f) C8; (g) C9. The y axis shows the annual median mixing ratio fraction of organic vapors for each carbon number, which means that for different organic vapors with the same carbon number, the sum of the mixing ratio fractions equals 1. The unprotonated formulae of organics vapors with multiple oxygens are labelled. In molecular formulas with the same number of carbons and oxygens, the hydrogen content in the organic vapors with the highest intensity is emphasized by bold and underlined formatting.

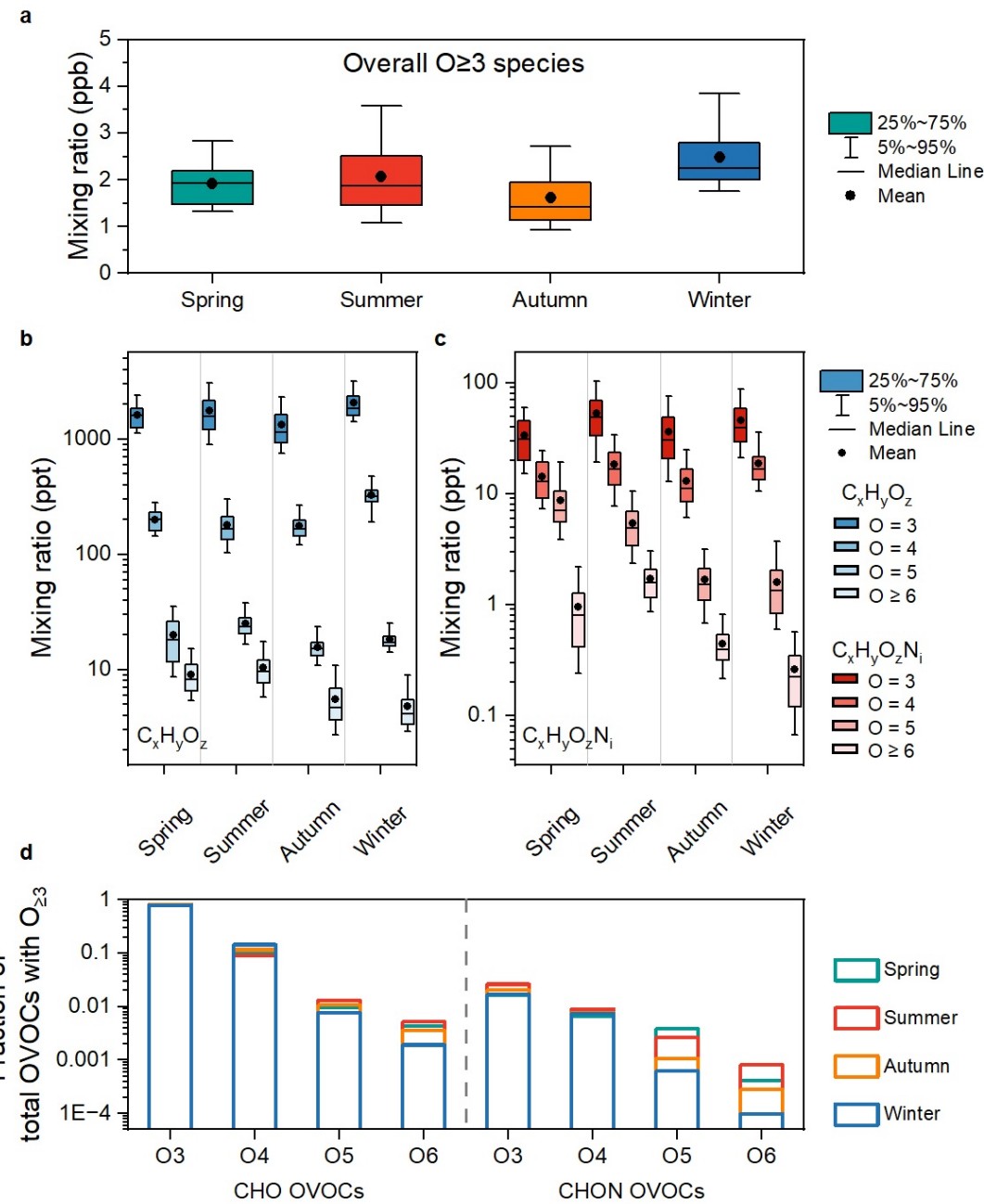

1161

Figure 5. Seasonal variations of organic vapors with multiple oxygens in urban
Beijing. (a) Total organic vapors with multiple oxygens. (b) $C_xH_yO_z$ with different
oxygens. (c) $C_xH_yO_zN_i$ with different oxygens. (d) Fractions of organic vapors
with different oxygens of total organic vapors with multiple oxygens.

1166

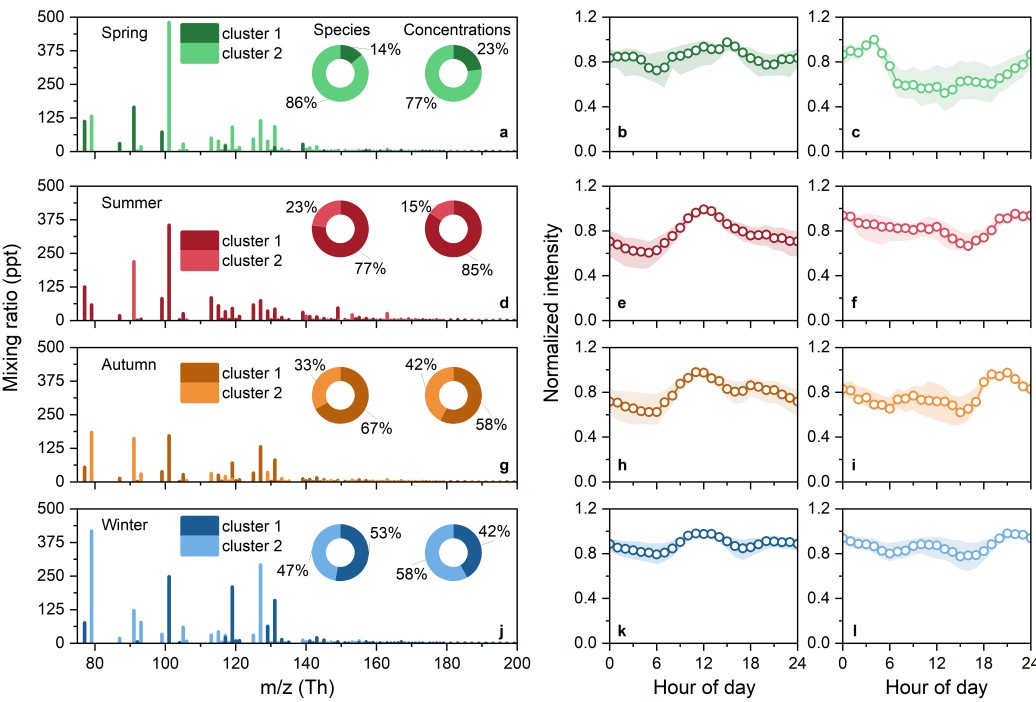

1167

Figure 6. Cluster results of organic vapors with multiple oxygens in four seasons. (a) – (c) Cluster results for spring. (a) Mass spectra of organic vapors with multiple oxygens in spring. Y axis is the median mixing ratio of each compound. Two different shades of colors are used to distinguish between two clusters. Two pie charts represent the distribution of species numbers and mixing ratios of organic vapors for two clusters. (b) Normalized median diurnal variation of cluster 1, daytime cluster. (c) Normalized median diurnal variation of cluster 2, nighttime cluster. The shaded areas in the graph (b) and (c) represent the 25th and 75th percentiles. (d) – (f) Cluster results for summer. (g) to (i) Cluster results for autumn. (j) – (l) Cluster results for winter.