# Peer review of "Molecular and seasonal characteristics of organic vapors in"

_EGUsphere, 2024_

## Author Comment (AC1)

**Responses to Community' Comments on Manuscript EGUSPHERE- 2024-1325**

(Molecular and seasonal characteristics of organic vapors in urban Beijing: insights from Vocus-PTR measurements)

Thanks for the comments. We have addressed them in the following paragraphs and made the corresponding changes in the revised manuscript. The comments are shown in blue italic text, followed by our responses. Changes in the revised manuscript are highlighted and presented as "quoted underlined text" in our responses.

*1. In Figure S3, the transmission efficiency of C8 aromatics (C8H11+) is greater than 1. Does the authors have any explanation for this?*

**Response:** We have modified the method of determining the sensitivity and recalculate the transmission efficiency.

Firstly, we used more compounds to determine the linearity. As mentioned in Table S2, we used 2 cylinders of calibration gas to calibrate the Vocus-PTR during different observation periods. Although the sensitivities of calibration gases varied across different observation periods, the relative sensitivities to toluene were comparable. We plot the sensitivities of the 2 cylinders of calibration gas together in Figure R1a. The y axis is the normalized sensitivity to toluene, and the x axis is their corresponding $k_{PTR}$. The black squares represent calibration gases from cylinder 1, and the black dots represent calibration gases from cylinder 2.

Then, we refitted the linearity using $C_7H_9^+$, $C_8H_{11}^+$, $C_9H_{13}^+$, $C_{10}H_9^+$, and $C_5H_9O_2^+$, with the result of the linear fit shown by the black line in the figure. The equation is $y = 0.43x+0.23$ with an $R^2$ of 0.87. Note that the sensitivity of toluene needs to be multiplied when using the equation. The species with gray labels have lower sensitivities due to the influence of transmission, so it is necessary to correct for the transmission efficiency.

Thirdly, we calculated the transmission efficiency based on these calibration gases, except for $C_5H_9^+$, $C_{10}H_{17}^+$ and $C_{11}H_{11}^+$, as shown in Figure R1b. The cut off is around 40, we have added this information in the main text. We revised this part in the main text.

For the new transmission efficiency, the average transmission efficiency of C8-C10 aromatics is slightly above 1, but this is reasonable within the margin of error.

[Figure]

Figure R1 (also shown as Figure S3 in the supporting information). Calibration results of mixed calibration gases. (a) The scatter plot of the sensitivities of mixed calibration gases and their $k_{PTR}$. The blue line is the linear fitting of $C_7H_9^+$, $C_8H_{11}^+$, $C_9H_{13}^+$, $C_{10}H_9^+$, and $C_5H_9O_2^+$, respectively. The error bar refers to standard deviation. The sensitivities of species with gray labels are affected by transmission. (b) The transmission efficiency of mixed calibration gases. The blue line is the fitted transmission efficiency curve based on that of mixed calibration gases. The error bar refers to standard deviation.

[Line 200 to 216] Figure S3a shows the measured sensitivities of mixed calibration gases and their corresponding $k_{PTR}$ values. The linear regression between kPTR and sensitivities was obtained based on sensitivities of $C_7H_9^+$, $C_8H_{11}^+$, $C_9H_{13}^+$, $C_{10}H_9^+$, and $C_5H_9O_2^+$ with an $R^2$ of 0.87. Sensitivities of other ions in mixed calibration gases may be influenced by transmission (ions labeled as gray) and fragmentation ($C_5H_9^+$, $C_{10}H_{17}^+$ and $C_{11}H_{11}^+$). The transmission efficiency of mixed calibration gases was calculated using sensitivities of mixed calibration gases, as shown in Figure S3b. The transmission efficiency of mixed calibration gases aligns well with the fitted transmission efficiency curve, except for $C_5H_9^+$, $C_{10}H_{17}^+$ and $C_{11}H_{11}^+$, which potentially experience fragmentation (fragmentation of measured ions are discussed below). For organic vapors without standards, their theoretical $k_{PTR}$ were used to constrain sensitivities, while for organic vapors with no theoretical $k_{PTR}$, an average $k_{PTR}$ of known species, $2.5 \times 10^{-9}$ $cm^3$ $molecule^{-1}$ $s^{-1}$ was used to constrain their sensitivities. The theoretical $k_{PTR}$ of organic vapors are from previous studies (Zhao and Zhang, 2004; Cappellin et al., 2012; Sekimoto et al., 2017).

*2. What were the Limit of detection (LoD) values of the VOCs containing more than 6 oxygen atoms?*

**Response:** We calculated the 1 min LODs for both calibrated and uncalibrated compounds using zero-gas background measurements taken every 2 hours during the observation periods (as shown in Figure R2). The LODs were calculated as 3 times the standard deviation of the zero-gas background divided by the obtained sensitivity. Very few formulae of VOCs containing more than 6 oxygen atoms were detected in this study, and we group them together with VOCs containing 6 oxygen atoms. The LODs for VOCs containing 6 or more oxygens is 0.06 ± 0.04 ppt.

[Figure]

Figure R2 (also shown as Figure S4 in the supplementary). Average limits of detection (1 min) for detected compounds. Different colors refer to different oxygen number of compounds, as labelled in legend.

---

## Author Comment (AC2)

**Responses to Reviewers' Comments on Manuscript EGUSPHERE- 2024-1325**

(Molecular and seasonal characteristics of organic vapors in urban Beijing: insights from Vocus-PTR measurements)

We have addressed each comment in the following paragraphs and made the corresponding changes in the revised manuscript. The reviewers' comments are shown in blue italic text, followed by our responses. Changes in the revised manuscript are highlighted and presented as "quoted underlined text" in our responses.

**Reviewer #1:**

*Scientific significance:*

*In my opinion the significance of this paper is good. It shows the differences of VOC emissions for different seasons and also analyzes the influence of day and night times on VOC emissions.*

*Scientific quality:*

*The scientific quality is ok, however the errors of the measurements need to be included and better presented.*

*Presentation quality:*

*The presentation quality is good but can be improved by explaining what the influence of OSc and DBE is. And why it is important for this paper. It is explained in detail how it is calculated (which in my opinion can also be placed in the supporting material part) but the influence on the atmosphere is not made clear enough.*

**Response**: We appreciate the comments. We have addressed the above general comments in detail in the following responses.

*General comments:*

*First of all, you need to clarify the term concentration. This term isn't used correctly in the whole paper. A concentration is defined as mass/volume what you measured with you Vocus is a mixing ratio (in this case a volume mixing ratio). You correctly used the unit ppt/ppb. However, to further clarify that you are talking about volume mixing-ratios you can either say it once and than say you will use ppb/ppt as a unit of volume mixing ratios or you can use.*

**Response**: The reviewer is correct. We have replaced the term "concentration" with "mixing ratio" throughout the manuscript.

*Introduction*

*You don't need to include all the different PTRs. Here just make clear what the advantage is (higher sensitivity and lower detection limits), how it is achieved ("incorporating radio frequency electric fields to focus ions") and the disadvantage (lighter ions are cut off to protect the detector from overloading). Another disadvantage compared to GC and other methods is that you can't be sure on the exact compound. You only get the information of the exact mass and from this you get information on a sum formular. However, you don't know anything about the functional groups etc. (no chance to differentiate between ketones and aldehydes).*

**Response**: Thanks for the suggestion. We have summarized the improvements in PTR and described them in one sentence. We also added their disadvantages, one being the cut off of lighter ions, and the other being the inability to distinguish the exact compound. The modifications in the main text are as follows:

[Line 78 to 85] Recent developments in the ion-molecule reactor (IMR) configuration have greatly increased sensitivities and concurrently lowered the limits of detection of PTR-MS by several orders of magnitude by incorporating radio frequency electric fields to focus ions (Breitenlechner et al., 2017; Krechmer et al., 2018; Reinecke et al., 2023). A consequential issue is that these advanced PTR-MS typically need to eliminate lighter ions to protect the detector from overload, and similar to traditional PTR-MS, they are incapable of obtaining molecular structure information.

*Why are you only comparing data from PTR, if you want to see higher oxidized compounds people used other methods. (Iodide-CIMS, nitrate-CIMS...). However I don't know if studies were conducted with those techniques in the past in this way. If there were studies, just include some examples.*

**Response**: Thanks for the suggestion. Yes, other CIMS (nitrate, iodide, bromide, and ammonium-CIMS) are usually used to study oxygenated organic molecules. We have included them in the Introduction. This paragraph in the main text is revised as follows:

[Line 86 to 101] These improvements have expanded the detection capabilities of PTR-MS, particularly for organic vapors with lower volatility and multiple oxygens ($\geq 3$) (Riva et al., 2019), which enables the simultaneous measurement of VOC precursors and their primary, secondary, and higher-level oxidation products using a single instrument (Li et al., 2020). Despite their low concentrations, these vapors may condense onto pre-existing aerosols and make a significant contribution to secondary aerosol growth and cloud condensation nuclei (Bianchi et al., 2019; Pospisilova et al., 2020; Nie et al., 2022). Organic vapors with multiple oxygens are likely to be simultaneously detected by other chemical ionization mass

spectrometry (CIMS), e.g., nitrate ($NO_3^-$), iodide ($I^-$), bromide ($Br^-$), and ammonium ($NH_4^+$) (Riva et al., 2019; Huang et al., 2021), which are widely used for measuring oxygenated organic compounds in the atmosphere (Bianchi et al., 2019; Ye et al., 2021; Huang et al., 2021). Therefore, using these improved PTR-MS can supplement our understanding of oxygenated organic vapors and facilitate the study of atmospheric chemical evolution of organics (Wang et al., 2020a).

**2.1.**

*How do you account for fragments, you seem to use a quite hard setting. What is your E/N? I know it is hard to calculate but you can find some help in this publication: Jensen, A., Koss, A. R., Hales, R., and de Gouw, J. A.: Measurements of VOCs in ambient air by Vocus PTR-TOF-MS: calibrations, instrument background corrections, and introducing a PTR Data Toolkit, Atmos. Meas. Tech., 16, 5261–5285, doi:10.5194/amt-16-5261-2023, 2023.*

**Response**: Thanks for the suggestion. The E/N in our study was 146.9 Td, which in case limited the formation of water clusters, promoted the simple reaction kinetics, but may lead to fragmentation. Here, we corrected the fragmentation, water cluster, and interferences for calibrated and uncalibrated species.

For α-pinene, we identified its fragments based on GC chromatograms. The Vocus-PTR was calibrated in GC mode before atmospheric observation. We tested a total of 4 species (shown in Figure R1a), including severely fragmented α-pinene. The spectrum of α-pinene is shown in the Figure R1b, with the main fragment being $C_6H_9^+$. We also investigated the potential interference of ion $C_{10}H_{19}O^+$. Since the correlation between $C_{10}H_{17}^+$ and $C_{10}H_{19}O^+$ was not strong (0.33 < r < 0.72) and the mixing ratio of $C_{10}H_{19}O^+$ was two orders of magnitude lower than $C_{10}H_{17}^+$, the impact of $C_{10}H_{19}O^+$ was not considered.

[Figure]

Figure R1. GC-Vocus results. (a) GC chromatogram of 4 species. (b) MS spectrum of α-pinene.

Several long-chain aldehydes and cycloalkanes may fragment on $C_5H_8H^+$, the ion typically attributed to isoprene in PTR-MS (Gueneron et al., 2015; Pfannerstill et al., 2023; Coggon et al., 2024). We corrected isoprene signals following an approach by Coggon et al. (2024). The correction is calculated as follows:

$$\text{m/z } 69.07_{\text{Corrected}} = S_{69.07} - S_{111.12+125.13} \cdot f_{69.07/(111.12+125.13)}$$

$S_{69.07}$ is the signal measured at $C_5H_9^+$. $S_{111.12+125.13}$ is the signal of the isoprene interferences, referring to $C_8H_{15}^+$ (m/z 111.12) and $C_9H_{17}^+$ (m/z 125.13), which are dehydrated products from octanal and nonanal, respectively. $f_{69.07/(111.12+125.13)}$ is determined from nighttime data (0:00-4:00) of each period. Similarly, acetaldehyde was corrected for ethanol fragments. We also checked the fragments and water cluster list in Pfannerstill et al. (2023) and Jensen et al. (2023). When the Pearson correlation coefficient r was greater than 0.95, we considered that the ions were fragments or water clusters of the parent ion.

We also tried to exclude the effects of unknown fragments and water clusters based on correlations of times series. Similar to Pfannerstill et al. (2023), any ion showing a correlation with another ion with $r^2 > 0.97$ (if chemical reasonable) was analyzed for possible water clustering or fragmentation effects and added up with its parent ion. The ions corrected are listed as follows: $C_2H_4N^+$ with water cluster $C_2H_6NO^+$, $C_3H_7O^+$ with water cluster $C_3H_9O_2^+$, $C_5H_9^+$ with fragment $C_5H_7^+$, $C_7H_9^+$ with fragment $C_7H_7^+$, $CH_4NO^+$ with water cluster $CH_6NO_2^+$, $C_2H_7O^+$ with water cluster $C_2H_9O_2^+$, $C_3H_3O_2^+$ with water cluster $C_3H_5O_3^+$, $C_4H_5O_2^+$ with water cluster $C_4H_7O_3^+$, $C_3H_5^+$ with fragment $C_3H_3^+$, $C_2H_5O^+$ with water cluster $C_2H_7O_2^+$, $C_2H_4NO^+$ with water cluster $C_2H_6NO_2^+$, $C_4H_5O_2^+$ with water cluster $C_4H_7O_3^+$, $C_3H_3O_3^+$ with water cluster $C_3H_5O_4^+$, $C_6H_6NO^+$ with water cluster $C_6H_8NO_2^+$, $C_8H_8NO_2^+$ with water cluster $C_8H_{10}NO_3^+$, $C_{10}H_{21}O^+$ with water cluster $C_{10}H_{23}O_2^+$, $C_9H_{13}O_3^+$ with water cluster $C_9H_{15}O_4^+$, $C_{10}H_{13}O_3^+$ with water cluster $C_{10}H_{15}O_4^+$, and $C_{14}H_{13}^+$ with water cluster $C_{14}H_{15}O^+$.

We acknowledge that this method cannot identify all fragments and clusters, and fragments and clusters may still be present in the measured VOCs and OVOCs. Further research is needed to explore the impact of fragments and clusters on the measurements, particularly concerning OVOCs with multiple oxygens.

We have added one paragraph in the main text to address the potential fragments and water clusters. See line 228 to 264 in the main text.

The fragmentation, water cluster, and interferences for calibrated and uncalibrated species were corrected. The ratio of the electric field strength (E) to the buffer gas number density (N) used in our study was 146.9 Td, and the gradient between BSQ skimmer 1 and skimmer 2 was 9.8 V, which in case limited the formation of water clusters, promoted the simple reaction kinetics, and improved the sensitivity, but may lead to stronger fragmentation. For α-pinene, we identified its fragments based on GC chromatograms. The Vocus-PTR was calibrated in GC mode before atmospheric measurement. A total of 4 species were tested in GC mode, including severely fragmented α-pinene. The spectrum of α-pinene showed that the main fragment was $C_6H_9^+$. Several long-chain aldehydes and cycloalkanes may fragment on $C_5H_8H^+$, the ion typically attributed to isoprene in PTR-MS (Gueneron et al., 2015; Pfannerstill et al., 2023a; Coggon et al., 2024). We corrected isoprene signals following an approach by Coggon et al. (2024). The correction was calculated as follows:

$$m/z\ 69.07_{Corrected} = S_{69.07} - S_{111.12+125.13} \cdot f_{69.07/(111.12+125.13)} \quad (1)$$

$S_{69.07}$ is the signal measured at $C_5H_9^+$. $S_{111.12+125.13}$ is the signal of the isoprene interferences, referring to $C_8H_{15}^+$ (m/z 111.12) and $C_9H_{17}^+$ (m/z 125.13), which are dehydrated products from octanal and nonanal, respectively. $f_{69.07/(111.12+125.13)}$ was determined from nighttime data (0:00-4:00) of each period. Similarly, acetaldehyde was corrected for ethanol fragments. We also checked the fragments and water cluster list in Pfannerstill et al. (2023a) and Jensen et al. (2023). When the Pearson correlation coefficient r is greater than 0.95, the ions were considered as fragments or water clusters of the parent ion. We also tried to exclude the effects of unknown fragments and water clusters based on correlations of times series. Similar to Pfannerstill et al. (2023a), any ion showing a correlation with another ion with $r^2 > 0.97$ (if chemical reasonable) was analyzed for possible water clustering or fragmentation effects and added up with its parent ion. The ions corrected are listed as follows: $C_2H_4N^+$ with water cluster $C_2H_6NO^+$, $C_3H_7O^+$ with water cluster $C_3H_9O_2^+$, $C_5H_9^+$ with fragment $C_5H_7^+$, $C_7H_9^+$ with fragment $C_7H_7^+$, $CH_4NO^+$ with water cluster $CH_6NO_2^+$, $C_2H_7O^+$ with water cluster $C_2H_9O_2^+$, $C_3H_3O_2^+$ with water cluster $C_3H_5O_3^+$, $C_4H_5O_2^+$ with water cluster $C_4H_7O_3^+$, $C_3H_5^+$ with fragment $C_3H_3^+$, $C_2H_5O^+$ with water cluster $C_2H_7O_2^+$, $C_2H_4NO^+$ with water cluster $C_2H_6NO_2^+$, $C_4H_5O_2^+$ with water cluster $C_4H_7O_3^+$, $C_3H_3O_3^+$ with water cluster $C_3H_5O_4^+$, $C_6H_6NO^+$ with water cluster $C_6H_8NO_2^+$, $C_8H_8NO_2^+$ with water cluster $C_8H_{10}NO_3^+$, $C_{10}H_{21}O^+$ with water cluster $C_{10}H_{23}O_2^+$, $C_9H_{13}O_3^+$ with water cluster $C_9H_{15}O_4^+$, $C_{10}H_{13}O_3^+$ with water cluster $C_{10}H_{15}O_4^+$, and $C_{14}H_{13}^+$ with water cluster $C_{14}H_{15}O^+$.

*Was a heating installed around your inletline and was it kept constant? If not, this might also explain your observation of less IVOCs and SVOCs in winter times.*

**Response**: The sampling tube was heated to 50 ± 5°C during the observation periods to lower the impacts on IVOCs and SVOCs. We have added this information in the main text.

*Were the meteorological parameters somehow included in your analysis?*

**Response**: We included the analysis of meteorological parameters into the analysis of diurnal variations across different clusters, and also made corresponding revisions in the main text.

[Line 488 to 495] Daytime clusters start to rise at 6:00-7:00 (6:00 for summer and 7:00 for other seasons), peak at 11:00-14:00 and then slowly decrease, following the diurnal variation of solar radiation (Li et al., 2023), ozone and temperature (Fig. S2). Figure S10 further demonstrates the dependence of daytime clusters on temperature. The mixing ratio of daytime clusters show an apparent increase in summer (when temperature is higher than 15 °C), which indicates that higher temperatures accompanied by an increase in solar radiation and ozone favors the formation of daytime clusters.

[Line 497 to 500] In summer, the vast majority of species (77%) exhibit daytime characteristics, with a mixing ratio percentage as high as 85%, which may be related to the strongest solar radiation (Li et al., 2023) and lowest NOx concentrations (Fig. S2).

[Line 504 to 505] The afternoon peak of daytime clusters in autumn and winter are accompanied by a decrease in mixing layer height (Li et al., 2023).

[Line 522 to 524] Nighttime clusters also show better consistency with $PM_{2.5}$ compared to daytime clusters (Fig. S2), which may be related to mixed sources.

*2.2.*

*Your cut off is above the mentioned 35 amu. This needs to be told here. In the supplementary information the equations of linearity and transmission curve would also be a nice add on. Additionally, I don't like the idea of using the mean of those three compounds (supplementary). I understand that you had to exclude the others due to your transmission curve. However, the error will be too low compared to the error that is expected if you only have 3 compounds with which you actually had to get the linearity alone. It is not always true that the offset is "0" which you claimed to be true. If you had used a softer setting you might not be able to detect compounds with a low k-rate. Therefore, the offset might even be negative. This error could be minimized by using compounds with higher and lower k-rates. Here however, all compounds*

*had nearly the same k-rate. Luckily the k-rates were the k-rates that were by default anyhow used for most of the compounds. I would suggest to at least make this fit with those three compounds and if the error is high you need to show this.*

**Response**: Thanks for the comments and suggestions. We have modified this part and describe it as follows:

Firstly, we used more compounds to determine the linearity. As mentioned in Table S2, we used 2 cylinders of calibration gas to calibrate the Vocus-PTR during different observation periods. Although the sensitivities of calibration gases varied across different observation periods, the relative sensitivities to toluene were comparable. We plot the sensitivities of the 2 cylinders of calibration gas together in Figure R2a. The y axis is the normalized sensitivity to toluene, and the x axis is their corresponding $k_{PTR}$. The black squares represent calibration gases from cylinder 1, and the black dots represents calibration gases from cylinder 2.

Then, we refitted the linearity using $C_7H_9^+$, $C_8H_{11}^+$, $C_9H_{13}^+$, $C_{10}H_9^+$, and $C_5H_9O_2^+$, with the result of the linear fit shown by the black line in the figure. The equation is y = 0.43x+0.23 with an $R^2$ of 0.87. Note that the sensitivity of toluene needs to be multiplied when using the equation. The offset is 0.23, not 0, which is also in accordance with the review's comment. The species with gray labels have lower sensitivities due to the influence of transmission, so it is necessary to correct for the transmission efficiency.

Thirdly, we calculated the transmission efficiency based on these calibration gases, as shown in Figure R2b. the cut off is around 40, we have added this information in the main text.

Lastly, we updated the whole measurement data using the new linearity and transmission efficiency.

[Figure]

[Figure]

Figure R2 (also shown as Figure S3 in the supplementary). Calibration results of mixed calibration gases. (a) The scatter plot of the sensitivities of mixed calibration gases and their $k_{PTR}$. The blue line is the linear fitting of $C_7H_9^+$, $C_8H_{11}^+$, $C_9H_{13}^+$, $C_{10}H_9^+$, and $C_5H_9O_2^+$, respectively. The error bar refers to standard deviation. The sensitivities of species with gray labels are affected by transmission. (b) The transmission efficiency of mixed calibration gases. The blue line is the fitted transmission efficiency curve based on that of mixed calibration gases. The error bar refers to standard deviation.

We have also revised the main text:

[Line 200 to 216] Figure S3a shows the measured sensitivities of mixed calibration gases and their corresponding $k_{PTR}$ values. The linear regression between kPTR and sensitivities was obtained based on sensitivities of $C_7H_9^+$, $C_8H_{11}^+$, $C_9H_{13}^+$, $C_{10}H_9^+$, and $C_5H_9O_2^+$ with an $R^2$ of 0.87. Sensitivities of other ions in mixed calibration gases may be influenced by transmission (ions labeled as gray) and fragmentation ($C_5H_9^+$, $C_{10}H_{17}^+$ and $C_{11}H_{11}^+$). The transmission efficiency of mixed calibration gases was calculated using sensitivities of mixed calibration gases, as shown in Figure S3b. The transmission efficiency of mixed calibration gases aligns well with the fitted transmission efficiency curve, except for $C_5H_9^+$, $C_{10}H_{17}^+$ and $C_{11}H_{11}^+$, which potentially experience fragmentation (fragmentation of measured ions are discussed below). For organic vapors without standards, their theoretical $k_{PTR}$ were used to constrain sensitivities, while for organic vapors with no theoretical $k_{PTR}$, an average $k_{PTR}$ of known species, $2.5 \times 10^{-9}$ $cm^3$ $molecule^{-1}$ $s^{-1}$ was used to constrain their sensitivities. The theoretical $k_{PTR}$ of organic vapors are from previous studies (Zhao and Zhang, 2004; Cappellin et al., 2012; Sekimoto et al., 2017).

*The fragmentation of C10H17+ would be nice to see, the most abundant fragment is C9H6+. (to have an idea on the fragmentation strength; however, this is only helpful not mandatory)*

**Response**: Thanks for the suggestion. We have checked and corrected the fragments of $C_{10}H_{17}^+$ based on GC chromatograms, and the fragmentation ratio was $52.8 \pm 10.6$ for $C_{10}H_{17}^+$. We have added one paragraph in the main text to address the potential fragments and water clusters. Please refer to the response to the comment "*How do you account for fragments...*" and see line 228 to 264 in the main text.

*Make clear why you use DBE and OSc. What do you expect and what does it say? If you answer this, your analysis part will be easier to understand.*

**Response**: Thanks for the suggestion. DBE represents the degree of unsaturation. The DBE of organic vapor with multiple oxygens is influenced by oxidation process and its precursors (if has). For example, aromatic VOCs have DBE values no smaller than 4, while aliphatic VOCs usually have DBE values smaller than 2 (Nie et al., 2022). For organic vapors with DBE between 2-3, they are likely oxidation products of aliphatic and aromatic VOCs (Nie et al., 2022). We also compare the OSc for organic vapors with different oxygens. For the same number of carbon atoms, organic vapors with a higher number of oxygen atoms exhibit a higher carbon oxidation state, which indicates a functionalization process (Kroll et al., 2011). We have revised and added discussions on DBE and OSc in the main text, and we also moved the calculation methods of DBE, OSc, and volatility to the Supporting Information.

[Line 371 to 379] Aromatic VOCs have DBE values no smaller than 4, while aliphatic VOCs usually have DBE values smaller than 2. For organic vapors with DBE between 2-3, they are likely oxidation products of aliphatic and aromatic VOCs (Wang et al., 2021b; Nie et al., 2022). For the same number of carbon atoms, organic vapors with a higher number of oxygen atoms exhibit a higher carbon oxidation state (as shown in Figure S5). Compared to organic vapors with 3 or 4 oxygen atoms, organic vapors with 5 or more oxygens have undergone more extensive atmospheric oxidation and functionalization processes (Kroll et al., 2011; Isaacman-Vanwertz et al., 2018).

*Line 220: there is an additional box in the text*
**Response**: Revised.

*Paragraph 328 ff*
*Isoprene is a bad example when ozone is present. In (https://amt.copernicus.org/articles/16/1179/2023/amt-16-1179-2023.pdf) it is described that oxidized compounds can fragment in the ion source of a PTR (also Vocus) and land on the exact mass as isoprene does. Therefore, the isoprene signal can be overestimated.*

**Response**: We have corrected isoprene signals following an approach by Coggon et al. (2024). We have added one paragraph in the main text to address the potential fragments and water clusters. Please refer to the response to the comment "*How do you account for fragments...*" and see line 228 to 264 in the main text.

*Paragraph 355 ff*
*Cold inlet line might also explain lower SVOC mixing ratios*

**Response**: Thanks for the reminder. The sampling tube was heated to $50 \pm 5°C$ during the observation periods to lower the impacts on IVOCs and SVOCs.

*"Day time cluster"*

*Do I understand correctly, all VOCs increase at 6 am? There should be seasonal changes (due to changing light conditions), or is there another source (e.g. traffic, factories?)*

**Response**: We appreciate the reviewer's suggestions. After carefully comparing the diurnal variations across different seasons, we found that in spring, autumn, and winter, the daytime clusters start to increase after 7:00 AM, whereas in summer, the increase begins after 6:00 AM. Another seasonal change is that the number and corresponding mixing ratios of species allocated to the daytime clusters vary in four seasons. In summer, the vast majority of species (77%) exhibit daytime characteristics, with a mixing ratio percentage as high as 85%, which may be related to the strongest solar radiation and lowest NOx concentrations. We also made revisions in the main text.

[Line 488 to 505] Daytime clusters start to rise at 6:00-7:00 (6:00 for summer and 7:00 for other seasons), peak at 11:00-14:00 and then slowly decrease, following the diurnal variation of solar radiation (Li et al., 2023), ozone and temperature (Fig. S2). Figure S10 further demonstrates the dependence of daytime clusters on temperature. The mixing ratio of daytime clusters show an apparent increase in summer (when temperature is higher than 15 °C), which indicates that higher temperatures accompanied by an increase in solar radiation and ozone favors the formation of daytime clusters. The number and corresponding mixing ratios of species allocated to the daytime clusters vary in four seasons. In summer, the vast majority of species (77%) exhibit daytime characteristics, with a mixing ratio percentage as high as 85%, which may be related to the strongest solar radiation (Li et al., 2023) and lowest NOx concentrations (Fig. S2). The contribution of daytime clusters in autumn is also significant, with 67% and 58% of the species and mixing ratios being accounted for. The noon peaks of daytime clusters in winter and spring are relatively less pronounced, with the species and mixing ratio day/night ratios also being comparatively lower. The afternoon peak of daytime clusters in autumn and winter are accompanied by a decrease in mixing layer height (Li et al., 2023).

*Is it possible to check for inversion layer and/or boundary layer. Especially, in winter this can decrease the efficiency of dilution. (meteorological data)*

**Response**: Thanks for the suggestion. Unfortunately, we did not measure boundary layer height. However, we checked and referenced the MLH data in Li et al. (2023) between December 2019 to August 2021 measured by a ceilometer. The sampling site in Li et al. (2023) was on the fifth floor of a building on the west campus of Beijing University of Chemical Technology, approximately 6.5 km away from our site. This MLH data covered the spring and summer time of our sampling period. We compared the other parameters measured in Li et al. with those in our study. Although the absolute values differ, the relative seasonal trends are consistent. We included MLH analysis in the main text.

[Line 477 to 478] The seasonal variations of OVOCs are partly caused by the variation of mixing layer height (Li et al., 2023), which is lowest in winter.

[Line 504 to 505] The afternoon peak of daytime clusters in autumn and winter are accompanied by a decrease in mixing layer height (Li et al., 2023).

*Line 457 style no "the" in front of winter*

**Response**: Revised.

*Line 459 style better: in winter in Beijing during the last few years*

**Response**: Revised.

*Line 466 just keep in mind that there will be fragments on the isoprene mass…*

**Response**: We have corrected isoprene signals following an approach by Coggon et al. (2024). We have added one paragraph in the main text to address the potential fragments and water clusters. Please refer to the response to the comment "*How do you account for fragments…*" and see line 228 to 264 in the main text.

*Supporting information*

*Figure S3 as mentioned above the compounds C8H11+, C9H13+ and C7H9+ would be needed to fit the k-rate to sensitivity line and not just the mean of all three slopes.*

**Response**: We have modified the method for determining the sensitivity. Please refer to the response to the comment "*Your cut off is above the mentioned 35 amu….*" and see line 200 to 216 in the main text.

*Figure S9 Which vmr is plotted here? The sum of all? What does this graph say? You see less when it's cold? (explain your figure in a few sentences)*

**Response**: The vmr plotted here is the sum of daytime clusters. The mixing ratio of daytime clusters show an apparent increase in summer (when temperature is higher than 15 Degree Celsius), which indicates that higher temperatures accompanied by an increase in solar radiation (Li et al., 2023) favors the formation of daytime clusters. We revised the figure caption and the main text.

[Line 492 to 495] The mixing ratio of daytime clusters show an apparent increase in summer (when temperature is higher than 15 °C), which indicates that higher temperatures accompanied by an increase in solar radiation and ozone favors the formation of daytime clusters.

*Figure S10 Which vmr is plotted against which? I assume it's sum of nighttime cluster (x-axis) against sum of CxHyO1-2 compounds?*

**Response**: It's the sum of nighttime clusters (x-axis) against the sum of the cluster 1 of CxHyO1-2 compounds. We have revised the caption.

*Table S2 under the table a "Benzene" is missing (1,3,-dichloro-) no ",", between 1,1-dichloro- and benzene (1,1-dichloro-benzene). Why are those compounds not included? They are quite heavy and it would definitely help to get a better idea on your k-rate to sensitivity plot and also in your transmission curve plot.*

**Response**: Thanks for the reminder. We tested some chlorine-containing compounds in the lab before the observation started, but our Vocus had very low sensitivity to these chlorine-containing compounds (as shown in Figure R3). Therefore, we did not include these compounds in the subsequent data processing. We also deleted this sentence to avoid misunderstanding and all species are used for instrument calibration in the revised table.

[Figure]

Figure R3. Sensitivities for chlorine-containing compounds and other compounds.

**References:**

Coggon, M. M., Stockwell, C. E., Claflin, M. S., Pfannerstill, E. Y., Xu, L., Gilman, J. B., Marcantonio, J., Cao, C., Bates, K., Gkatzelis, G. I., Lamplugh, A., Katz, E. F., Arata, C., Apel, E. C., Hornbrook, R. S., Piel, F., Majluf, F., Blake, D. R., Wisthaler, A., Canagaratna, M., Lerner, B. M., Goldstein, A. H., Mak, J. E., and Warneke, C.: Identifying and correcting interferences to PTR-ToF-MS measurements of isoprene and other urban volatile organic compounds, Atmos. Meas. Tech., 17, 801-825, 10.5194/amt-17-801-2024, 2024.

Gueneron, M., Erickson, M. H., VanderSchelden, G. S., and Jobson, B. T.: PTR-MS fragmentation patterns of gasoline hydrocarbons, Int. J. Mass Spectrom., 379, 97-109, 10.1016/j.ijms.2015.01.001, 2015.

Jensen, A. R., Koss, A. R., Hales, R. B., and de Gouw, J. A.: Measurements of volatile organic compounds in ambient air by gas-chromatography and real-time Vocus PTR-TOF-MS: calibrations, instrument background corrections, and introducing a PTR Data Toolkit, Atmos. Meas. Tech., 16, 5261-5285, 10.5194/amt-16-5261-2023, 2023.

Kroll, J. H., Donahue, N. M., Jimenez, J. L., Kessler, S. H., Canagaratna, M. R., Wilson, K. R., Altieri, K. E., Mazzoleni, L. R., Wozniak, A. S., Bluhm, H., Mysak, E. R., Smith, J. D., Kolb, C. E., and Worsnop, D. R.: Carbon oxidation state as a metric for describing the chemistry of atmospheric organic aerosol, Nature Chemistry, 3, 133-139, 10.1038/nchem.948, 2011.

Li, X., Chen, Y., Li, Y., Cai, R., Li, Y., Deng, C., Wu, J., Yan, C., Cheng, H., Liu, Y., Kulmala, M., Hao, J., Smith, J. N., and Jiang, J.: Seasonal variations in composition and sources of atmospheric ultrafine particles in urban Beijing based on near-continuous measurements, Atmos. Chem. Phys., 23, 14801-14812, 10.5194/acp-23-14801-2023, 2023.

Nie, W., Yan, C., Huang, D. D., Wang, Z., Liu, Y., Qiao, X., Guo, Y., Tian, L., Zheng, P., Xu, Z., Li, Y., Xu, Z., Qi, X., Sun, P., Wang, J., Zheng, F., Li, X., Yin, R., Dallenbach, K. R., Bianchi, F., Petäjä, T., Zhang, Y., Wang, M., Schervish, M., Wang, S., Qiao, L., Wang, Q., Zhou, M., Wang, H., Yu, C., Yao, D., Guo, H., Ye, P., Lee, S., Li, Y. J., Liu, Y., Chi, X., Kerminen, V.-M., Ehn, M., Donahue, N. M., Wang, T., Huang, C., Kulmala, M., Worsnop, D., Jiang, J., and Ding, A.: Secondary organic aerosol formed by condensing anthropogenic vapours over China's megacities, Nature Geoscience, 15, 255-261, 10.1038/s41561-022-00922-5, 2022.

Pfannerstill, E. Y., Arata, C., Zhu, Q., Schulze, B. C., Woods, R., Seinfeld, J. H., Bucholtz, A., Cohen, R. C., and Goldstein, A. H.: Volatile organic compound fluxes in the agricultural San Joaquin Valley – spatial distribution, source attribution, and inventory comparison, Atmos. Chem. Phys., 23, 12753-12780, 10.5194/acp-23-12753-2023, 2023.

---

## Author Comment (AC3)

**Responses to Reviewers' Comments on Manuscript EGUSPHERE- 2024-1325**

(Molecular and seasonal characteristics of organic vapors in urban Beijing: insights from Vocus-PTR measurements)

We appreciate the reviewers' comments and believe that our responses have improved this manuscript. We have addressed each comment in the following paragraphs and made the corresponding changes in the revised manuscript. The reviewers' comments are shown in blue italic text, followed by our responses. Changes in the revised manuscript are highlighted and presented as "quoted underlined text" in our responses.

**Reviewer #2:**

*This paper presents an analysis of highly oxygenated molecules measured by a Vocus PTR-ToF-MS in Beijing for one year. The authors present a seasonal analysis of their concentrations in addition to a cluster analysis which highlights which times of day certain types of highly oxygenated VOCs are present. They present relevant properties such as DBE, number of oxygens, and volatility.*

*While I believe that the data is interesting and should be published, the paper and analysis requires major edits before it can be considered for publication in ACP. Most importantly, the authors should improve their quantification techniques so they are more applicable to the molecules in question and include a more thorough discussion of limitations and uncertainties. The paper is also lacking proper justification for studying these highly oxygenated molecules. I am left with key contextual questions such as (1) What fraction of the measured concentration are these highly oxygenated molecules? (2) What fraction of the calculated OH reactivity or SOA formation potential are these highly oxygenated molecules? (3) Why should we focus on them? A very quick google scholar search of 'highly oxygenated molecules' reveals that there is more research on these molecules and their importance in SOA formation that suggested in the introduction. I am not an expert on this topic, so I would like to see more justification for studying these species with such low concentrations. In addition to these technical and context-related concerns, there are many grammatical errors in the paper, so it should be carefully edited with that in mind, as well.*

*For the main reasons discussed in the prior paragraph, and after reviewing my general and specific comments below, I think the paper should be reconsidered after major revisions. I do*

*encourage the authors to strongly consider my suggestions as well as the other reviewers suggestions for improving the paper and re-submit once they are addressed.*

**Response**: We appreciate the comments. We have addressed the above general comments in detail in the following responses.

*Abstract:*

*You switched from past to present tense a few times. Please stay consistent throughout the abstract and consider reporting your findings in the past tense in the abstract.*

**Response**: Thanks. We have revised the tense throughout the main text. Actions that occurred in the past use the past tense, while statements of fact (even if they are past phenomena) use the present tense.

*Introduction:*

*There is too much discussion on different PTR techniques for a paper whose results aren't related to method development. Keep discussion on PTR methods in the introduction to one paragraph maximum, maybe moving some of the discussion on PTR to the methods.*

**Response**: Thanks for your suggestion. We have shortened the introduction of the PTR instruments and kept it within one paragraph, while moving some of the explanations of the PTR principles to the methods section. This paragraph in the main text has been changed to:

[Line 66 to 85] Instrumental advances have allowed for improving the understanding of the compositions and variations of VOCs at the molecular level, especially for oxygenated VOCs (OVOCs). Gas chromatography or multidimensional gas chromatography coupled with mass spectrometry is the most commonly used technology for VOC measurement, capable of detecting major non-methane hydrocarbons and select OVOCs (Lewis et al., 2000; Xu et al., 2003; Noziere et al., 2015). Proton Transfer Reaction-Mass Spectrometry (PTR-MS) enables real-time detection of VOCs without pre-concentration and separation, greatly enriching the molecular understanding of OVOCs due to its high sensitivity to oxygen-containing compounds (Hansel et al., 1995; De Gouw and Warneke, 2007; Yuan et al., 2017). Hundreds of OVOCs are detected and characterized in different areas using PTR-MS, e.g. urban (Wu et al., 2020), suburban (He et al., 2022), and forest areas (Pugliese et al., 2023). Recent developments in the ion-molecule reactor (IMR) configuration have greatly increased sensitivities and concurrently lowered the limits of detection of PTR-MS by several orders of magnitude by incorporating radio frequency electric fields to focus ions (Breitenlechner et al.,

2017; Krechmer et al., 2018; Reinecke et al., 2023). A consequential issue is that these advanced PTR-MS typically need to eliminate lighter ions to protect the detector from overload, and similar to traditional PTR-MS, they are incapable of obtaining molecular structure information.

*Consider adding more background on oxygenated + highly oxygenated VOCs and why studying them is needed – this would frame the results of the paper more effectively. Specifically, I believe there should be more explanation in the introduction on why you are studying 'organic vapors with low mixing ratios.'*

**Response**: Thanks for the suggestion. We have added more background on oxygenated organic molecules and provided an explanation of how these organic vapors with low mixing ratios impact the atmosphere. This paragraph in the main text is revised as follows:

[Line 86 to 101] These improvements have expanded the detection capabilities of PTR-MS, particularly for organic vapors with lower volatility and multiple oxygens (≥3) (Riva et al., 2019), which enables the simultaneous measurement of VOC precursors and their primary, secondary, and higher-level oxidation products using a single instrument (Li et al., 2020). Despite their low concentrations, these vapors may condense on pre-existing aerosols and make a significant contribution to secondary aerosol growth and cloud condensation nuclei (Bianchi et al., 2019; Pospisilova et al., 2020; Nie et al., 2022). Organic vapors with multiple oxygens are likely to be simultaneously detected by other chemical ionization mass spectrometry (CIMS), e.g., nitrate ($NO_3^-$), iodide ($I^-$), bromide ($Br^-$), and ammonium ($NH_4^+$) (Riva et al., 2019; Huang et al., 2021), which are widely used for measuring oxygenated organic compounds in the atmosphere (Bianchi et al., 2019; Ye et al., 2021; Huang et al., 2021). Therefore, using these improved PTR-MS can supplement our understanding of oxygenated organic vapors and facilitate the study of atmospheric chemical evolution of organics (Wang et al., 2020a).

*There is far more recent published literature on urban VOCs than you suggest. See many recent papers published by Karl, Coggon, Pfannerstill, Gkatzelis, Acton, etc. etc.*

**Response**: Thanks for the suggestion. We have added more references and corresponding introduction in this paragraph. Yes, there are a number of published studies on urban VOCs, and we are focused on the observations from the improved PTRs. The sentences added to the main text are shown below:

[Line 121 to 122] Several studies have carried out measurements in urban air using these improved PTR-MS.

[Line 123 to 126] Coggon et al. (2024) evaluated the fragmentation and interferences of a series of urban VOCs. Pfannerstill et al. (2023 and 2024) measured hundreds of VOCs to calculate their emission fluxes in Los Angeles

*Methods:*

*For a results section focused almost exclusively on highly oxygenated molecules, the methods section is lacking discussion of their quantification. How feasible is it to calibrate your highly oxygenated VOCs using the sensitivity of three aromatic VOCs? Have you attempted to quantify any highly oxygenated VOCs or at least determine their fragmentation ratios?*

**Response**: Thanks for the suggestion. We agree and have made a series of improvements.

**(1) Quantification method**

We agree with the reviewer that using the mean sensitivity of 3 compounds to get the linearity is inappropriate. We have modified this part and describe it as follows:

Firstly, we used more compounds to determine the linearity. As mentioned in Table S2, we used 2 cylinders of calibration gas to calibrate the Vocus-PTR during different observation periods. Although the sensitivities of calibration gases varied across different observation periods, the relative sensitivities to toluene were comparable. We plot the sensitivities of the 2 cylinders of calibration gas together in Figure R1a. The y axis is the normalized sensitivity to toluene, and the x axis is their corresponding $k_{PTR}$. The black squares represent calibration gases from cylinder 1, and the black dots represents calibration gases from cylinder 2.

Then, we refitted the linearity using $C_7H_9^+$, $C_8H_{11}^+$, $C_9H_{13}^+$, $C_{10}H_9^+$, and $C_5H_9O_2^+$, with the result of the linear fitting shown by the black line in the figure. The equation is $y = 0.43x+0.23$ with an $R^2$ of 0.87. Note that the sensitivity of toluene needs to be multiplied when using the equation. The species with gray labels have lower sensitivities due to the influence of transmission, so it is necessary to correct for the transmission efficiency.

Thirdly, we calculated the transmission efficiency based on these calibration gases, except for $C_5H_9^+$, $C_{10}H_{17}^+$ and $C_{11}H_{11}^+$, as shown in Figure R1b. The cut off is around 40, we have added this information in the main text.

Lastly, we updated the whole measurement data using the new linearity and transmission efficiency.

[Figure]

Figure R1 (also shown as Figure S3 in the SI). Calibration results of mixed calibration gases. (a) The scatter plot of the sensitivities of mixed calibration gases and their $k_{PTR}$. The blue line is the linear fitting of $C_7H_9^+$, $C_8H_{11}^+$, $C_9H_{13}^+$, $C_{10}H_9^+$, and $C_5H_9O_2^+$, respectively. The error bar refers to standard deviation. The sensitivities of species with gray labels are affected by transmission. (b) The transmission efficiency of mixed calibration gases. The blue line is the fitted transmission efficiency curve based on that of mixed calibration gases. The error bar refers to standard deviation.

We also calculated the 1 min LODs for both calibrated and uncalibrated compounds using zero-gas background measurements taken every 2 hours during the observation periods, as shown in Figure R2. The LODs were calculated as 3 times the standard deviation of the zero-gas background divided by the obtained sensitivity. Data below the LODs were excluded from further analysis.

[Figure]

Figure R2 (also shown as Figure S4 in the supplementary). Average limits of detection (1 min) for detected compounds. Different colors refer to different oxygen number of compounds, as labelled in legend.

We have modified this part in the main text. See line 200 to 227 in the main text.

Figure S3a shows the measured sensitivities of mixed calibration gases and their corresponding $k_{PTR}$ values. The linear regression between kPTR and sensitivities was obtained based on sensitivities of $C_7H_9^+$, $C_8H_{11}^+$, $C_9H_{13}^+$, $C_{10}H_9^+$, and $C_5H_9O_2^+$ with an $R^2$ of 0.87. Sensitivities of other ions in mixed calibration gases may be influenced by transmission (ions labeled as gray) and fragmentation ($C_5H_9^+$, $C_{10}H_{17}^+$ and $C_{11}H_{11}^+$). The transmission efficiency of mixed calibration gases was calculated using sensitivities of mixed calibration gases, as shown in Figure S3b. The transmission efficiency of mixed calibration gases aligns well with the fitted transmission efficiency curve, except for $C_5H_9^+$, $C_{10}H_{17}^+$ and $C_{11}H_{11}^+$, which potentially experience fragmentation (fragmentation of measured ions are discussed below). For organic vapors without standards, their theoretical $k_{PTR}$ were used to constrain sensitivities, while for organic vapors with no theoretical $k_{PTR}$, an average $k_{PTR}$ of known species, $2.5\times10\text{-}9$ cm$^3$ molecule$^{-1}$ s$^{-1}$ was used to constrain their sensitivities. The theoretical $k_{PTR}$ of organic vapors are from previous studies (Zhao and Zhang, 2004; Cappellin et al., 2012; Sekimoto et al., 2017). Average limits of detection (LODs, 1 min) of the measured compounds were determined using zero-gas background measurements taken every 2 hours during the observation periods, as shown in Figure S4. The LODs were calculated as 3 times the standard deviation of the zero-gas background divided by the obtained sensitivity. The LODs show a correlation with masses; as masses increase, instrument backgrounds decrease, leading to lower LODs. This trend was

observed for species with different oxygen content, with LODs around 0.03 ± 0.03 pptv at m/z 200. Note that LODs in this study are one-minute averages, with raw 1-second data averaged to 1 minute before Tofware analysis as mentioned before, which may account for the lower LODs compared to those in Jensen et al. (2023). Data below the LODs were excluded from further analysis.

**(2) Correction for fragmentations, water clusters, and interferences**

Here, we corrected the fragmentation, water cluster, and interferences for calibrated and uncalibrated species.

For α-pinene, we identified its fragments based on GC chromatograms. The Vocus-PTR was calibrated in GC mode before the formal observation. We tested a total of 4 species (shown in Figure R3a), including severely fragmented α-pinene. The spectrum of α-pinene is shown in the Figure R3b, with the main fragment being $C_6H_9^+$. We also investigated the potential interference of ion $C_{10}H_{19}O^+$. Since the correlation between $C_{10}H_{17}^+$ and $C_{10}H_{19}O^+$ was not strong (0.33 < r < 0.72) and the mixing ratio of $C_{10}H_{19}O^+$ was two orders of magnitude lower than $C_{10}H_{17}^+$, the impact of $C_{10}H_{19}O^+$ was not considered.

[Figure]

Figure R3. GC-Vocus results. (a) GC chromatogram of 4 species. (b) MS spectrum of α-pinene.

Several long-chain aldehydes and cycloalkanes may fragment on $C_5H_8H^+$, the ion typically attributed to isoprene in PTR-MS (Gueneron et al., 2015; Pfannerstill et al., 2023; Coggon et al., 2024). We corrected isoprene signals following an approach by Coggon et al. (2024). The correction is calculated as follows:

$$\text{m/z } 69.07_{\text{Corrected}} = S_{69.07} - S_{111.12+125.13} \cdot f_{69.07/(111.12+125.13)}$$

$S_{69.07}$ is the signal measured at $C_5H_9^+$. $S_{111.12+125.13}$ is the signal of the isoprene interferences, referring to $C_8H_{15}^+$ (m/z 111.12) and $C_9H_{17}^+$ (m/z 125.13), which are dehydrated products from octanal and nonanal, respectively. $f_{69.07/(111.12+125.13)}$ is determined from nighttime data (0:00-

4:00) of each period. Similarly, acetaldehyde was corrected for ethanol fragments. We also checked the fragments and water cluster list in Pfannerstill et al. (2023) and Jensen et al. (2023). When the Pearson correlation coefficient r was greater than 0.95, we considered that the ions were fragments or water clusters of the parent ion.

We also tried to exclude the effects of unknown fragments and water clusters based on correlations of times series. Similar to Pfannerstill et al. (2023), any ion showing a correlation with another ion with $r^2 > 0.97$ (if chemical reasonable) was analyzed for possible water clustering or fragmentation effects and added up with its parent ion. The ions corrected are listed as follows: $C_2H_4N^+$ with water cluster $C_2H_6NO^+$, $C_3H_7O^+$ with water cluster $C_3H_9O_2^+$, $C_5H_9^+$ with fragment $C_5H_7^+$, $C_7H_9^+$ with fragment $C_7H_7^+$, $CH_4NO^+$ with water cluster $CH_6NO_2^+$, $C_2H_7O^+$ with water cluster $C_2H_9O_2^+$, $C_3H_3O_2^+$ with water cluster $C_3H_5O_3^+$, $C_4H_5O_2^+$ with water cluster $C_4H_7O_3^+$, $C_3H_5^+$ with fragment $C_3H_3^+$, $C_2H_5O^+$ with water cluster $C_2H_7O_2^+$, $C_2H_4NO^+$ with water cluster $C_2H_6NO_2^+$, $C_4H_5O_2^+$ with water cluster $C_4H_7O_3^+$, $C_3H_3O_3^+$ with water cluster $C_3H_5O_4^+$, $C_6H_6NO^+$ with water cluster $C_6H_8NO_2^+$, $C_8H_8NO_2^+$ with water cluster $C_8H_{10}NO_3^+$, $C_{10}H_{21}O^+$ with water cluster $C_{10}H_{23}O_2^+$, $C_9H_{13}O_3^+$ with water cluster $C_9H_{15}O_4^+$, $C_{10}H_{13}O_3^+$ with water cluster $C_{10}H_{15}O_4^+$, and $C_{14}H_{13}^+$ with water cluster $C_{14}H_{15}O^+$.

We have added one paragraph in the main text to address the potential fragments and water clusters. See line 228 to 264 in the main text.

The fragmentation, water cluster, and interferences for calibrated and uncalibrated species were corrected. The ratio of the electric field strength (E) to the buffer gas number density (N) used in our study was 146.9 Td, and the gradient between BSQ skimmer 1 and skimmer 2 was 9.8 V, which in case limited the formation of water clusters, promoted the simple reaction kinetics, and improved the sensitivity, but may lead to stronger fragmentation. For α-pinene, we identified its fragments based on GC chromatograms. The Vocus-PTR was calibrated in GC mode before atmospheric measurement. A total of 4 species were tested in GC mode, including severely fragmented α-pinene. The spectrum of α-pinene showed that the main fragment was $C_6H_9^+$. Several long-chain aldehydes and cycloalkanes may fragment on $C_5H_8H^+$, the ion typically attributed to isoprene in PTR-MS (Gueneron et al., 2015; Pfannerstill et al., 2023a; Coggon et al., 2024). We corrected isoprene signals following an approach by Coggon et al. (2024). The correction was calculated as follows:

$$m/z\ 69.07_{Corrected} = S_{69.07} - S_{111.12+125.13} \cdot f_{69.07/(111.12+125.13)} \quad (1)$$

$S_{69.07}$ is the signal measured at $C_5H_9^+$. $S_{111.12+125.13}$ is the signal of the isoprene interferences, referring to $C_8H_{15}^+$ (m/z 111.12) and $C_9H_{17}^+$ (m/z 125.13), which are dehydrated products from octanal and nonanal, respectively. $f_{69.07/(111.12+125.13)}$ was determined from nighttime data (0:00-4:00) of each period. Similarly, acetaldehyde was corrected for ethanol fragments. We also checked the fragments and water cluster list in Pfannerstill et al. (2023a) and Jensen et al. (2023). When the Pearson correlation coefficient r is greater than 0.95, the ions were considered as fragments or water clusters of the parent ion. We also tried to exclude the effects of unknown fragments and water clusters based on correlations of times series. Similar to Pfannerstill et al. (2023a), any ion showing a correlation with another ion with $r^2 > 0.97$ (if chemical reasonable) was analyzed for possible water clustering or fragmentation effects and added up with its parent ion. The ions corrected are listed as follows: $C_2H_4N^+$ with water cluster $C_2H_6NO^+$, $C_3H_7O^+$ with water cluster $C_3H_9O_2^+$, $C_5H_9^+$ with fragment $C_5H_7^+$, $C_7H_9^+$ with fragment $C_7H_7^+$, $CH_4NO^+$ with water cluster $CH_6NO_2^+$, $C_2H_7O^+$ with water cluster $C_2H_9O_2^+$, $C_3H_3O_2^+$ with water cluster $C_3H_5O_3^+$, $C_4H_5O_2^+$ with water cluster $C_4H_7O_3^+$, $C_3H_5^+$ with fragment $C_3H_3^+$, $C_2H_5O^+$ with water cluster $C_2H_7O_2^+$, $C_2H_4NO^+$ with water cluster $C_2H_6NO_2^+$, $C_4H_5O_2^+$ with water cluster $C_4H_7O_3^+$, $C_3H_3O_3^+$ with water cluster $C_3H_5O_4^+$, $C_6H_6NO^+$ with water cluster $C_6H_8NO_2^+$, $C_8H_8NO_2^+$ with water cluster $C_8H_{10}NO_3^+$, $C_{10}H_{21}O^+$ with water cluster $C_{10}H_{23}O_2^+$, $C_9H_{13}O_3^+$ with water cluster $C_9H_{15}O_4^+$, $C_{10}H_{13}O_3^+$ with water cluster $C_{10}H_{15}O_4^+$, and $C_{14}H_{13}^+$ with water cluster $C_{14}H_{15}O^+$.

**(3) Discuss on uncertainties for uncalibrated compounds, including OVOCs with multiple oxygens**

We have added one paragraph in the main text.

[Line 265 to 279] Here, we discuss the uncertainties of quantification for calibrated and uncalibrated compounds. The uncertainty of calibrated ions ranges from 2% to 16% determined from the standard deviations of the fast calibrations during the measurement periods. The semi-quantification was conducted for uncalibrated compounds with their sensitivities constrained by $k_{PTR}$ linear relationship and transmission efficiency. The uncertainty of these uncalibrated compounds arising from linear fitting and transmission efficiency fitting is 20% using Monte Carlo simulation. Additionally, undetermined fragmentations and water clusters also contribute to the uncertainty, though we identified some potential fragments and water clusters through the strength of correlations as previously indicated. We acknowledge that this method cannot identify all fragments and clusters, and fragments and clusters may still be present in the measured VOCs and OVOCs. Further research is needed to explore the impact of fragments and clusters on the measurements, particularly concerning OVOCs with multiple oxygens.

*Do you expect any of your reported formulae are fragments or water clusters? The Vocus is prone to high levels of fragmentation, and this should be investigated for this dataset and discussed. See methods of Pfannerstill et al. 2023 ACP for a list of possible water clusters and fragments to look out for (https://doi.org/10.5194/acp-23-12753-2023).*

**Response**: Thanks for the suggestion. We have checked and corrected the fragments and water clusters and added one paragraph in the main text to address this. Please refer to the response to the comment "*For a results section focused almost exclusively on highly oxygenated molecules…*" or see line 228 to 264 in the main text.

*There needs to be a discussion about the limits of detection for these highly oxygenated molecules.*

**Response**: Thanks for the suggestion. We have included the LODs in this study. Please refer to the response to the comment "*For a results section focused almost exclusively on highly oxygenated molecules…*" or see line 216 to 227 in the main text.

*Results:*

*Regarding section 3.1 on identifying VOC formulae, I believe more quality assurance should be included here since this is a central part of your results. Was the peak identification performed manually or automatically in Tofware or other program? Did you set a detection threshold for identifying peaks (i.e., what's your limit of detection)? What maximum mass error was allowed for identifying peaks? How confident are you about formulae identifications, especially at high molecular weights where you might run into ambiguous assignments?*

**Response**: Thanks for the suggestion. We have included more information on this part in the main text.

[Line 181 to 194] Data analysis of Vocus-PTR mass spectra, including mass calibration, baseline subtraction, and high-resolution peak fitting was conducted using Tofware (v3.2.3, Tofwerk AG and Aerodyne Research Inc.) within the Igor Pro 8 platform (WaveMetrics, OR, USA). The ambient mass spectra were averaged over 1 min for subsequent processing in Tofware. The peaklist used for high-resolution peak fitting was manually made based on mass spectra of both clean days ($PM_{2.5} < 75$ μg/m$^3$) and polluted days ($PM_{2.5} \geq 75$ μg/m$^3$). The maximum mass error allowed for identifying peaks is 5-10 ppm, which is consistent of the error of mass calibration. When there are multiple options of formulas meeting the error limit under, especially at high molecular weights, a peak with oxygen numbers ≤ 8 and carbon

numbers ≤ 20, and lower degree of unsaturation were selected; otherwise, the peak would be classified as unknown peak. The maximum peak area residual for each unit mass resolution is 5%. Subsequent analysis was performed in MATLAB R2022a (The MathWorks Inc., USA).

We also included limit of detection of each measured compound and set it as the detection threshold for each compound. Please refer to the response to the comment "*For a results section focused almost exclusively on highly oxygenated molecules…*" or see line 216 to 227 in the main text.

*I think there should be more discussion on the context of the highly oxygenated molecules. You define this group of compounds to focus on as VOCs with 3 or more oxygen atoms. But what fraction of the total measured concentration does this group comprise? What fraction of the total OH reactivity and/or SOA formation potential? The paper would be much stronger with this added context and motivation.*

**Response**: We thank the reviewer for the suggestions. These organic vapors with multiple oxygens account for 4% of the mixing ratios of total detected VOCs. We added this information in the main text.

[Line 386 to 388] The annual median mixing ratio of measured organic vapors with multiple oxygens in median ± standard deviation is 2.0 ppb ± 1.0 ppb, accounting for 4% of the total $C_xH_yO_z$ and $C_xH_yO_zN_i$ mixing ratios.

We also included calculations and discussion of condensation growth rates and OH reactivities in our analysis. Please see details in the main text:

[Line 280 to 291] The condensational growth rates contributed by detected organic vapors were simulated using a kinetic partitioning method, as detailed in Li et al. (2024b). For comparison, the condensational growth rates of low volatile and extremely low volatile organic compounds measured by nitrate-CIMS were also simulated (Li et al., 2024b). The OH reactivities of detected organic vapors were calculated, and the rate constants are from Data S1 in Pfannerstill et al. (2024) and Table S4 in Wu et al. (2020). For species with unreported rate constants, we calculated the OH reactivities for hydrocarbons and OVOCs using the reported median rate constants of hydrocarbons and OVOCs, respectively.

[Line 406 to 419] Though the contribution of the measured IVOCs and SVOCs to the overall VOC mixing ratio is low, their contribution to the condensational growth rates is non-negligible, which may influence the growth of new particles (Ehn et al., 2014), SOA formation (Jimenez

et al., 2009), and haze (Nie et al., 2022). The condensational growth rates of total organic vapors are calculated, including extremely low, low, and semi volatile organic compounds detected by nitrate-CIMS and I/SVOCs detected by Vocus-PTR. The contribution to the condensational growth rate from I/SVOCs detected by Vocus-PTR increases with particle size and decreases with temperature. For 8 nm particles, the contribution of SVOCs detected by Vocus-PTR is 9%, while IVOCs contribute 1%. For 40 nm particles, the contribution of SVOCs increases to 13%, and IVOCs rise to 4%. At sub-zero temperatures for 8 nm particles, the SVOC contribution detected by Vocus-PTR can reach up to 21%, with IVOCs contributing 10%.

[Line 439 to 450] Measured molecular formulae may react with OH radicals, contributing to OH reactivity. The calculated OH reactivity of organic vapors with multiple oxygens account for 6% of the total detected VOCs, with an average annual value of 1.2 $s^{-1}$. Previous studies show differences between measured and calculated or modeled OH reactivity (Hansen et al., 2014), and unmeasured species from photochemical oxidation likely explain this gap (Ferracci et al., 2018). Therefore, the OH reactivity contributed by detected organic vapors with multiple oxygens in this study may potentially reduce this gap, thereby improve the accuracy of diagnosis of sensitivity regimes for ozone formation (Wang et al., 2024). Using Vocus-PTR has the potential to simultaneously measure both precursors and multi-generational oxygenated products, which is beneficial for studying the evolution process of organic compounds in the atmosphere.

*Specific comments:*
*Abstract:*
*Line 28: Change 'compositions' to 'composition'*
**Response**: Revised.

*Introduction:*
*Line 113-116: Add references*
**Response**: Revised.

*Lin 137: Delete 'at molecular level'*
**Response**: Revised.

*Methods:*

*Line 144: Delete 'traffic' or 'roads'*

**Response**: Revised.

*Line 163-164: How regularly was the filter changed?*

**Response**: The filter was changed every 7 days to prevent the orifice from clogging. The data within 30 minutes after membrane replacement was excluded. We also added this information in the main text.

[Line 167 to 170] The sampling tube was heated to 50 ± 5°C during the measurement. A regularly replaced Teflon filter (every 7 days) was used in front of the sampling line to prevent the orifice from clogging. The data within 30 minutes after membrane replacement was excluded.

*Concerns about inlet design - Was your inlet heated and did you test any flow rates besides 3 LPM to assess adsorption of sticky VOCs and/or IVOC/SVOCs? How do you think the inlet impacts your measurements of lower volatility highly oxygenated VOCs?*

**Response**: Thanks for your suggestion. The sampling tube was heated to 50 ± 5°C during the observation periods to lower the impact on IVOCs and SVOCs. As for the flow rates of sampling, we tested the flow rates for monoethanolamine (MEA), a highly viscous VOC. A constant flow of zero air was used to purge the MEA permeation tube to generate MEA gas, while another flow of zero air was used as dilution gas. The two gases were first mixed and then split into two streams. One stream entered the Vocus-PTR (about 150 sccm), while the other was the excess flow. Figure R4 shows the variation of MEA intensity with the dilution gas flow rates. A good linearity ($R^2$ = 0.9979) was observed for flow rates ranging from 2.25 to 4.5 L/min. This indicates that a flow rate of 3 LPM was able to reduce wall losses; otherwise, we could not obtain such a good linearity.

[Figure]

Figure R4. The variation of MEA intensity with the dilution gas flow rates.

*Line 170: 5.6 km feels a bit far away, was this location usually downwind or upwind of your measuring site?*

**Response**: We appreciate the reviewer's comments. We have corrected the distance to 3.6 km. The previous value of 5.6 km was a typographical error, and we apologize for that. We have updated the map of the observation site, as shown in Figure S1. Additionally, we plot the wind rose charts of the observation periods, as shown below. The Wanliu station is located to the southwest of our observation site and is typically either upwind or downwind of the observation site.

[Figure]

Figure R5. Wind rose plot of the observation periods.

*Line 184: What BSQ voltage was used? What was the lowest m/z you were able to detect at 100% transmission with that voltage? This m/z depends on the specific BSQ voltage used.*

**Response**: The BSQ voltage used was 275 V, with a cut off m/z of about 40. The lowest m/z at 100% transmission was around 80. We have modified the method of determining the sensitivity and recalculated the transmission efficiency. Please refer to the comment "*For a results section*

*focused almost exclusively on highly oxygenated molecules…*" or see line 200 to 216 in the main text.

*Line 190-193: Saying that you used the average of three VOC sensitivities would be more honest. Before seeing the SI figure, I interpreted this as if you used those three VOCs in the fit of sensitivity versus kPTR. Since there doesn't appear to be a linear relationship between kPTR and sensitivity here, maybe try a different approach. Consider using an average of your well behaved calibrants (i.e., those outside of the BSQ filtering range that do not fragment or cluster to a large extent and the sum of monoterpene parent and major fragment [C10H17+C6H9])*

**Response**: Thanks for the comments and suggestions. We have modified the method of determining the sensitivity. Please refer to the comment "*For a results section focused almost exclusively on highly oxygenated molecules…*" or see line 200 to 216 in the main text.

*Line 195: Note that Figure S3b is the transmission of VOCs through the BSQ, correct? Was this transmission curve used calibrate VOCs in the m/z region impacted by the BSQ?*

**Response**: Figure S3b is the overall transmission efficiency of VOCs in Vocus-PTR, which included the transmission of BSQ, ToF and other transmission unit. The calculation method for transmission efficiency is the same as described in Krechmer et al. (2018):

Firstly, we determined the relationship between sensitivity and $k_{PTR}$ and found a linear correlation above the m/z of toluene (as shown in Figure R2 and Figure S3a). The lower sensitivities for VOCs below 93 Th ($C_7H_9^+$) were affected by BSQ.

Then, the ratios between the measured and calculated sensitivities for VOCs below 93 Th were defined as transmission efficiency and plotted in Figure R2 and Figure S3b.

Lastly, we fitted the transmission using an exponential fit. For VOCs below 93 Th, isoprene was excluded when fitting the transmission due to its fragmentation.

*Did you correct for ToF transmission?*

**Response**: This question is related to the previous question. Please see the response to the previous question.

*Line 202: Wrong unit – need molecule in the denominator.*

**Response**: Revised.

*Line 235: 'Square Euclidean' Instead of 'Sqeuclidean'?*

**Response**: Revised.

*Please include uncertainty estimates for calibrated and non-calibrated VOCs.*

**Response**: Thanks for the suggestion. We have included the uncertainty of calibrated and non-calibrated VOCs in the main text. Please refer to the comment "*For a results section focused almost exclusively on highly oxygenated molecules…*" or see line 265 to 279 in the main text.

*Please include your E/N ratio and see the supplement of Coggon et al. 2024 AMT (https://doi.org/10.5194/amt-17-801-2024) for a discussion on how other parameters (i.e., skimmer gradient) can also impact fragmentation in the vocus.*

**Response**: Thanks for the suggestion. The E/N used in our study was 146.9 Td, and the gradient between BSQ skimmer 1 and skimmer 2 was 9.8 V, which in case limited the formation of water clusters, promoted the simple reaction kinetics, and improved the sensitivity, but let to stronger fragmentation (Coggon et al., 2024). We have included this part in the main text and corrected for potential fragments. Please refer to the comment "*For a results section focused almost exclusively on highly oxygenated molecules…*" or see line 229 to 264 in the main text.

*Results:*

*Line 242: I think you mean number of molecular formulae instead of 'number of organics'?*

**Response**: Yes, we mean number of formulae and we revised this.

*Line 250-252: Please indicate what time resolution you are referring to for these concentrations.*

**Response**: The time resolution of these concentrations is 1 min. We revised this sentence as: [Line 321 to 325] The mixing ratios of organic vapors vary substantially in urban Beijing, ranging from 0.01 parts per trillion (ppt) to 10 parts per billion (ppb) in volume under a time resolution of 1 min, with many species detected at sub-ppt levels notably (Fig. 1d).

*Line 258: Consider saying 'identified' instead of 'discovered'*

**Response**: Revised.

*Line 268: Replace 'individually' with 'individual'*

**Response**: Revised.

*Line 297: Change 'organic vapors species' to 'organic species' or something else*

**Response**: Revised.

*Line 309: too many significant figures, consider rounding to 2800, same with '2352' in line 312.*

**Response**: Revised.

*Line 328-329: I don't see evidence to support this claim.*

*Line 332-333: Do you have evidence that these formulas are the oxidation products you think they might be (e.g., GC-MS)? With Vocus, you can only measure the formula which could have multiple isomers. Please correct the grammar here, too.*

**Response**: Thanks for the comments. Since these two comments are similar, we combine them in one response. We agree with the reviewer that Vocus can only measure the formula which could have multiple isomers. Unfortunately, the GC was only used for calibration of very limited species. Here we would like to say that we could observe these formulae in the atmosphere, and they are could potentially be oxidation products as previous reported, but confirming through GC and other methods is still needed. We revised the wording of this entire paragraph.

[Line 420 to 440] The molecular formulae of the measured organic vapors with multiple oxygens are displayed in the mass spectra, categorized by carbon numbers ranging from 2-11 (Fig. 4 and Table S3). Many of the formulae are reported as oxidation products of various VOC precursors in previous studies. Take isoprene as an example, detected formulae are reported as various oxidation products of isoprene, including $C_5H_{10}O_3$ and subsequent oxidation products in C5 species, e.g., $C_5H_8O_6$, $C_5H_9NO_4$, etc. (Wennberg et al., 2018). For several C4 species, such as $C_4H_7NO_4$, $C_4H_4O_3$, etc., they are reported as oxidation products of two additional important oxidation products of isoprene, methacrolein (MACR) and methyl vinyl ketone (MVK). We also see formulae reported as oxidation products of precursors such as benzene (C6) (Priestley et al., 2021), alkyl-substituted benzenes (C7-C9) (Pan and Wang, 2014; Wang et al., 2020c; Cheng et al., 2021), and monoterpenes (C10) (Rolletter et al., 2019). Besides, we can also detect some organic vapors with relatively low DBE (≤3), which may originate from the oxidation of aliphatic precursors. For example, $C_5H_8O_4$ observed are reported as one of the oxidation products of C5 aldehyde, the photolysis of which release OH radicals. This mechanism may explain the source gap of OH radicals between simulations and observations in low nitrogen oxide and high VOCs regimes (Yang et al., 2024). Note that these species may

be oxidation products as reported by previous studies; however, confirming this would require additional techniques such as GC.

*Line 344: I don't think you have evidence to support the claim that this may explain the missing source of OH radicals. Also, what do you mean missing source?*

**Response**: Similar as the previous response, we would like to express that we observed $C_5H_8O_4$, and it is reported as one of the oxidation products of C5 aldehyde. Here, the term "missing source" refers to the underestimation of OH concentration in model simulations compared to direct observations in low nitrogen oxide and high VOCs regimes; this underestimated portion is the missing source. We revised this sentence to:

[Line 434 to 438] For example, $C_5H_8O_4$ observed are reported as one of the oxidation products of C5 aldehyde, the photolysis of which release OH radicals. This mechanism may explain the source gap of OH radicals between simulations and observations in low nitrogen oxide and high VOCs regimes (Yang et al., 2024).

*Line 349: I think grammar needs to be adjusted here. This phrase is unclear '...supplement the missing VOCs when calculating OH reactivity...'*

**Response**: Revised.

*Line 352-354: Is this rate constant reasonable? You seem to have arbitrarily chosen a rate constant that's on the order of many terpenes, which are considered to react very quickly with OH. And again, what do you mean by missing OH reactivity? Was that measured here? Need more context and especially need proper evidence. Please consider using a more methodical approach for estimating what the kOH should be for each formula and scaling each concentration by each kOH to get your total calculated OH reactivity and report how important these highly oxygenated molecules are relative to the other VOCs are on this scale. I am having a hard time contextualizing these low-concentration highly oxygenated molecules.*

**Response**: Thanks for the comment and suggestion. We recalculated the OH reactivity using reported rate constants, taken from Data S1 in Pfannerstill et al. (2024) and Table S4 in Wu et al. (2020). For species with unreported rate constants, we calculate the OH reactivities for hydrocarbons and OVOCs using the reported median rate constants of hydrocarbons and OVOCs, respectively. Missing OH reactivity refers to the gap between measured and calculated or modeled OH reactivity. We have revised this part in the main text. Please refer to the

response to the comment "*I think there should be more discussion on the context of the highly oxygenated molecules ...*" and see line 441 to 452 in the main text.

*Line 357: Reduce number of significant figures*

**Response**: Revised.

*Line 370: 'Significant' - if you don't mean statistically significant, change word to 'substantially' or something similar.*

**Response**: Thanks. We changed the word to 'substantially'.

*Line 377: Statistically significant? If so, please indicate p value?*

**Response**: We did not mean statistically significant, and we have changed the word to 'substantially'.

*Line 418-419: I feel like you may be lacking evidence here. How do you know they are more influenced by secondary sources or that they even have primary sources.*

**Response**: We agree with the reviewer and have deleted this sentence in the main text.

*Line 465: Overserved? --> observed*

**Response**: Revised.

*Line 466: Did you investigate fragmentation onto isoprene's parent mass? See Coggon et al. 2024 AMT https://doi.org/10.5194/amt-17-801-2024*

**Response**: We have corrected isoprene signals following an approach by Coggon et al. (2024). We have added one paragraph in the main text to address the potential fragments and water clusters. Please refer to the comment "*For a results section focused almost exclusively on highly oxygenated molecules...*" or see line 228 to 264 in the main text.

*Line 503: Do you mean up to 230 m/z were observed in this study? Did you look into siloxanes like D4, D5, etc.?*

**Response**: Yes, we mean up to 230 m/z were observed in this study. However, the peakfitting was not continuous after m/z 201, and analysis only focused on m/z 201 and before. For D4 and D5, we did see them in the mass spectra, but we mainly focused on the organic compounds

containing C, H, O, and N atoms. D5 was used as a mass calibrant for mass calibration in Tofware due to its single gaussian peak and high intensity.

*Line 508: Do you mean 'urban areas'?*
**Response**: Yes, revised.

*Line 511: don't need to say both dominant and main*
**Response**: Revised.

*Line 513: urban areas*
**Response**: Revised.

*Line 532: change discovered to identified or something else.*
**Response**: Revised.

*Line 533: content instead of contents*
**Response**: Revised.

*Line 537: urban area to urban areas*
**Response**: Revised.

*Line 562: change measurement to measurements*
**Response**: Revised.

*Figure 1: Add labels to pie charts in (b) and (c). In the caption, (b) should be number of organic formulae I think?*
**Response**: Revised.

*Figure 2: To make this figure more clear, please add label to legend in (a) saying something like 'number of oxygen atoms.' Please add labels to pie charts in so you can see clearly what each one is plotting, i.e., 'concentration of CHO species'.*
**Response**: Revised.

*Figure 3: Final sentence of caption – change 'Y axials' to 'Y axes'*

**Response**: Revised.

*Figure 4: Please indicate if the molecular formula in figure 4 are protonated or not. If they are the protonated ion formulas, please indicate they are charged.*

**Response**: The molecular formula in figure 4 are not protonated. We added this information in the figure caption.

*Figure 5: (a) I can barely see the dot distribution to the left of the box plot for 'spring'. (d) Y axis label should read 'Fraction of' instead of 'Fraction to'*

**Response**: Revised.

*Figure 6: Please add a legend for cluster 1 and cluster 2 in the plots to make this clearer. Shading of percentiles is too light in c, f, h, I, k, and l.*

**Response**: Revised.

**References:**

Coggon, M. M., Stockwell, C. E., Claflin, M. S., Pfannerstill, E. Y., Xu, L., Gilman, J. B., Marcantonio, J., Cao, C., Bates, K., Gkatzelis, G. I., Lamplugh, A., Katz, E. F., Arata, C., Apel, E. C., Hornbrook, R. S., Piel, F., Majluf, F., Blake, D. R., Wisthaler, A., Canagaratna, M., Lerner, B. M., Goldstein, A. H., Mak, J. E., and Warneke, C.: Identifying and correcting interferences to PTR-ToF-MS measurements of isoprene and other urban volatile organic compounds, Atmos. Meas. Tech., 17, 801-825, 10.5194/amt-17-801-2024, 2024.

Gueneron, M., Erickson, M. H., VanderSchelden, G. S., and Jobson, B. T.: PTR-MS fragmentation patterns of gasoline hydrocarbons, Int. J. Mass Spectrom., 379, 97-109, 10.1016/j.ijms.2015.01.001, 2015.

Jensen, A. R., Koss, A. R., Hales, R. B., and de Gouw, J. A.: Measurements of volatile organic compounds in ambient air by gas-chromatography and real-time Vocus PTR-TOF-MS: calibrations, instrument background corrections, and introducing a PTR Data Toolkit, Atmos. Meas. Tech., 16, 5261-5285, 10.5194/amt-16-5261-2023, 2023.

Krechmer, J., Lopez-Hilfiker, F., Koss, A., Hutterli, M., Stoermer, C., Deming, B., Kimmel, J., Warneke, C., Holzinger, R., Jayne, J., Worsnop, D., Fuhrer, K., Gonin, M., and de Gouw, J.: Evaluation of a New Reagent-Ion Source and Focusing Ion-Molecule Reactor for Use in Proton-Transfer-Reaction Mass Spectrometry, Anal Chem, 90, 12011-12018, 10.1021/acs.analchem.8b02641, 2018.

Pfannerstill, E. Y., Arata, C., Zhu, Q., Schulze, B. C., Woods, R., Seinfeld, J. H., Bucholtz, A., Cohen, R. C., and Goldstein, A. H.: Volatile organic compound fluxes in the agricultural San Joaquin Valley – spatial distribution, source attribution, and inventory comparison, Atmos. Chem. Phys., 23, 12753-12780, 10.5194/acp-23-12753-2023, 2023.

Pfannerstill, E. Y., Arata, C., Zhu, Q., Schulze, B. C., Ward, R., Woods, R., Harkins, C., Schwantes, R. H., Seinfeld, J. H., Bucholtz, A., Cohen, R. C., and Goldstein, A. H.: Temperature-dependent emissions dominate aerosol and ozone formation in Los Angeles, Science, 384, 1324-1329, doi:10.1126/science.adg8204, 2024.

Wu, C., Wang, C., Wang, S., Wang, W., Yuan, B., Qi, J., Wang, B., Wang, H., Wang, C., Song, W., Wang, X., Hu, W., Lou, S., Ye, C., Peng, Y., Wang, Z., Huangfu, Y., Xie, Y., Zhu, M., Zheng, J., Wang, X., Jiang, B., Zhang, Z., and Shao, M.: Measurement report: Important contributions of oxygenated compounds to emissions and chemistry of volatile organic compounds in urban air, Atmos. Chem. Phys., 20, 14769-14785, 10.5194/acp-20-14769-2020, 2020.

---

## Author Response (AR2)

**Responses to Reviewers' Comments on Manuscript EGUSPHERE- 2024-1325**

(Molecular and seasonal characteristics of organic vapors in urban Beijing: insights from Vocus-PTR measurements)

Dear editor,

We appreciate both the reviewers' and the editor's comments and believe that our responses have improved this manuscript. We have addressed each comment in the following paragraphs and made the corresponding changes in the revised manuscript. The reviewers' comments are shown in blue italic text, followed by our responses. Changes in the revised manuscript are made using the track changes mode.

Regarding the data points below the LOD, we have redone the analysis with these points included. The m/z that were always below the LOD have been removed in the previous version.

Thank you again for your and the reviewers' time and efforts. We look forward to your continued feedback.

Best regards,
Zhaojin on behalf of the authors

**Reviewer #1:**
*Lines 255-264: It is in general important to see that you accounted for those fragments and water clusters but this is very hard to read. Maybe you can make a table and/or put it in the supplement. Another way could be to just use bullet points.*
*E.g. "We accounted for several fragments and water clusters, specified in table X in the supplement."*
**Response:** Thanks. We added a table in the supplement to clearly present the parent ions and their corresponding fragments or water cluster which we accounted for.

*Supplement:*
*In general, I would like to understand the plots in the supplement without having to read the whole paper to understand what is plotted against what. You don't need to go into too much detail why you see what you see. But the general idea of what I see on a graph should be self-*

*explanatory.*

*The following figures are lacking with respect to being easily understandable without a surrounding text. Take the following comments as suggestions, where to improve.*

*Figure S11: specify the x and y axes. As the caption alone is not helping.*

*Figure S12: What does 1 and 2 mean? The day and nighttime clusters?*

*Figure S13: What does 0-4 stand for? The different seasons? If so, why does it differ to table S1?*

*Figure S15: Maybe repeat the color code in the caption*

**Response:** Thanks for the suggestions. We have revised the figures in the supplement based on the reviewers' comments and carefully reviewed the entire SI, making adjustments to unclear figures and captions.

**Reviewer #2:**

*Grammatical edit suggestions:*

*Line 35: change 'spectrometry' to 'spectrometer'*

*Line 43: delete 'of'*

*Line 45: change 'generate' to 'generated'*

*Line 47: change 'are' to 'were'*

*Line 61: change 'radial' to 'radical'*

*Line 76: change 'are' to 'have been'*

**Response:** Revised.

*Other comments:*

*Line 226: "Data below the LODs were excluded from further analysis." Does this mean you deleted all data points where the mixing ratio was < LOD? Data below LOD should not be removed, as it will bias results high when you are calculating statistics on the dataset.*

**Response:** Thanks. We have already removed the m/z that were always below LODs, and we updated the whole analysis with data points below LODs included.

*Line 268, 292: Not sure if this is considered 'semi-quantification', I think it's just 'quantification'. Most studies I am familiar with refer to this as quantification with an uncertainty of about 20-50% due to the range of potential kPTR values possible. Maybe choosing particular kPTR values could cause a lower uncertainty for certain compounds.*

**Response:** Thanks. We changed 'semi-quantification' to 'quantification'.